# Dynamic centriolar localization of Polo and Centrobin in early mitosis primes centrosome asymmetry

Emmanuel Gallaud[1☉¤a], Anjana Ramdas Nair[1☉¤b], Nicole Horsley[2], Arnaud Monnard[1,2], Priyanka Singh[1¤c], Tri Thanh Pham[2¤d], David Salvador Garcia[1¤e], Alexia Ferrand[1], Clemens Cabernard[2]*

1 Biozentrum, University of Basel, Klingelbergstrasse, Basel, Switzerland, 2 Department of Biology, University of Washington, Life Science Building, Seattle, Washington State, United States of America

☉ These authors contributed equally to this work.
¤a Current address: CNRS, Université Rennes, Institut de Génétique et Développement de Rennes, Rennes, France
¤b Current address: New York University Abu Dhabi, Abu Dhabi, United Arab Emirates
¤c Current address: Department of Bioscience & Bioengineering, Indian Institute of Technology Jodhpur, Karwar, India
¤d Current address: Department of Biology, School of Sciences and Humanities, Nazarbayev University, Nur-Sultan, Republic of Kazakhstan
¤e Current address: Division of Cell Biology, MRC Laboratory of Molecular Biology, Cambridge, United Kingdom
* ccabern@uw.edu

**Data Availability Statement:** Source data files are available for all data presented in the main and supplementary figures.

## Abstract

Centrosomes, the main microtubule organizing centers (MTOCs) of metazoan cells, contain an older "mother" and a younger "daughter" centriole. Stem cells either inherit the mother or daughter-centriole-containing centrosome, providing a possible mechanism for biased delivery of cell fate determinants. However, the mechanisms regulating centrosome asymmetry and biased centrosome segregation are unclear. Using 3D-structured illumination microscopy (3D-SIM) and live-cell imaging, we show in fly neural stem cells (neuroblasts) that the mitotic kinase Polo and its centriolar protein substrate Centrobin (Cnb) accumulate on the daughter centriole during mitosis, thereby generating molecularly distinct mother and daughter centrioles before interphase. Cnb's asymmetric localization, potentially involving a direct relocalization mechanism, is regulated by Polo-mediated phosphorylation, whereas Polo's daughter centriole enrichment requires both Wdr62 and Cnb. Based on optogenetic protein mislocalization experiments, we propose that the establishment of centriole asymmetry in mitosis primes biased interphase MTOC activity, necessary for correct spindle orientation.

## Introduction

Centrosomes consist of a pair of centrioles, embedded in structured layers of pericentriolar material (PCM) [1]. Within the centrosome, centrioles differ by age and replicate once during

**Funding:** This work was supported by the Swiss National Science Foundation (SNSF; PP00P3_159318 to CC), the National Institutes of Health (NIH; 1R01GM126029-01 to CC) and start-up funds from the University of Washington (CC). E.G was supported with an EMBO long-term postdoctoral fellowship (ALTF 378-2015). Stocks obtained from the Bloomington Drosophila Stock Center (NIH P40OD018537) and the Vienna Drosophila Resource Center (VDRC) were used in this study. The funders had no role in study design, data collection and analysis, decision to publish, or preparation of the manuscript.

**Competing interests:** The authors have declared that no competing interests exist.

**Abbreviations:** Asl, Asterless; AEL, after egg laying; Cnb, Centrobin; FRAP, fluorescence recovery after photobleaching; GMC, ganglion mother cell; iLID, improved light-induced dimer; LOV, Light-Oxygen-Voltage; MTOC, microtubule organizing center; PACT, Pericentrin-AKAP-450 containing targeting; PCM, pericentriolar material; PCNT, Pericentrin; Plp, PCNT-like protein; RNAi, RNA interference; S/N, Signal/Noise; SspB, Stringent starvation protein B; SsrA, 10SA RNA; Wdr62, WD40 repeat protein 62; YFP, Yellow Fluorescent Protein; 3D-SIM, 3D structured illuminated microscopy.

the cell cycle. For instance, metazoan cells in G1 of the cell cycle usually contain 2 centrioles joined by a flexible linker. Both centrioles duplicate in S phase, forming a younger "daughter" centriole around a central cartwheel at a right angle to the existing older "mother" centriole. As cells enter mitosis, the linker between the 2 pairs of centrioles is severed, causing them to move apart and form a bipolar spindle. Both sibling cells inherit one centrosome, each consisting of an older "mother" and younger "daughter" centriole. As cells exit mitosis, the mother and daughter centrioles disengage, and the replication cycle starts again in G1 [2–4]. Many metazoan cells segregate centrosomes nonrandomly, ensuring that the oldest centriole will always be retained by a particular daughter cell, providing a possible mechanism to determine or influence cell fate decisions [5]. For instance, vertebrate neural stem cells and *Drosophila* male germline stem cells both retain the centrosome with the oldest mother centriole (mother centrosome hereafter) [6,7], whereas *Drosophila* female germline or neural stem cells, called neuroblasts, inherit the centrosome with the daughter centriole (referring to the previous cell cycle; daughter centrosome hereafter) [8–10]. Centrosome asymmetry is also manifested in the unequal clustering of proteins or mRNA [11–13].

In *Drosophila* male germline or neural stem cells, asymmetric centrosome function mediates spindle orientation [7,14]. Correct spindle orientation is necessary for cell-cycle progression, stem cell homeostasis and differentiation [15–17]. However, the mechanisms establishing functional centrosome asymmetry are incompletely understood. Furthermore, how centrosome asymmetry affects biased centrosome segregation remains elusive.

Here, we use *Drosophila* neuroblasts to investigate the spatiotemporal mechanisms underlying the establishment of centrosome asymmetry in vivo. Neuroblast centrosomes are highly asymmetric in interphase: one centrosome forms an active MTOC, while its sibling remains inactive until entry into mitosis [9,14,18]. The active interphase MTOC contains the daughter centriole from the previous cell cycle, identifiable with the orthologue of the human daughter-centriole-specific protein Cnb (Cnb⁺) [8]. This biased MTOC activity is regulated by the mitotic kinase Polo (Plk1 in vertebrates). Polo phosphorylates Cnb, necessary to maintain an active MTOC, tethering the daughter-centriole-containing centrosome to the apical interphase cortex (the apical centrosome hereafter) [19]. Apical centrosome tethering predetermines the alignment of the mitotic spindle along the intrinsic apical-basal polarity axis. Furthermore, this cortical association ensures that the daughter centrosome is inherited by the self-renewing neuroblast [8,19]. Polo localization on the apical centrosome is maintained by the microcephaly associated protein Wdr62 [20]. The mother centrosome, separating from the daughter centrosome in interphase, down-regulates Polo and MTOC activity through Pericentrin (PCNT)-like protein (Plp) and Bld10 (Cep135 in vertebrates) [21,22]. The lack of MTOC activity prevents the mother centrosome from engaging with the apical cell cortex; it randomly migrates through the cytoplasm until centrosome maturation in prophase establishes a second MTOC near the basal cortex (called the basal centrosome hereafter), ensuring its segregation into the differentiating ganglion mother cell (GMC). Later in mitosis, the mother centrosome also accumulates Cnb [9,14,18,21].

Although several centrosomal proteins have been described to be enriched on either the mother or daughter centrosome in *Drosophila* interphase neuroblasts [8,20,23] or human cells [24–26], it is unknown when and how centrosomes acquire their unique molecular identity to determine biased MTOC activity and thus correct spindle orientation. Here, we show that centrosome asymmetry is primed in early mitosis by dynamically enriching Polo and Cnb on the younger daughter centriole, while selectively retaining Plp on the mother centriole. We further show that priming centrosome asymmetry in mitosis is necessary to establish molecularly distinct centrosomes, asymmetric MTOC activity and centrosome positioning.

## Results

### Neuroblast centriole duplication starts in interphase and completes in mitosis

To determine the onset of centrosome asymmetry establishment in larval neuroblast, we first investigated the centriole replication cycle. In vertebrate cells, centrioles replicate in interphase and convert to functional centrosomes during mitosis (reviewed in the work by Fu, Hagan, and Glover; the work by Conduit, Wainman, and Raff; and the work by Nigg and Stearns [2,4,27]), but it is unclear whether this also applies to fly neuroblasts. We used 3D structured illumination microscopy (3D-SIM), which has approximately twice the spatial resolution of standard confocal microscopy, and stained third instar neuroblasts with known centriolar and centrosomal markers. For all the 3D-SIM experiments, the cell-cycle stages were determined based on the organization of the microtubule network and cell shape (S1A Fig). We used Asl in conjunction with Sas-6 to determine the onset of cartwheel duplication and centriole conversion during the neuroblast cell cycle (S1B Fig). Consistent with previous reports [1,28–30], we found that Sas-6 was localized to the centriolar cartwheel whereas Asterless (Asl) surrounded the centriolar wall (S1C Fig). Asl has been shown to extend from the core centriolar region into the adjacent PCM and sequentially loads onto the new centriole during centriole-to-centrosome conversion (also referred to as mitotic centriole conversion), a mechanism generating a centriole-duplication and PCM-assembly competent centrosome [28,31,32]. Apical and basal interphase neuroblast centrosomes contained 2 Sas-6$^+$ cartwheels but only one Asl$^+$ centriole (S1C Fig; yellow arrowhead). During prophase, Asl gradually appeared around the second cartwheel to form a pair of fully matured centrioles. In telophase, centrioles seemed to lose their orthogonal conformation, possibly because of disengagement, before migration. Cartwheels started to duplicate in late telophase, manifested in the appearance of a third Sas-6 positive cartwheel (blue arrowhead in S1C Fig). Based on these data, we conclude that in third instar larval neuroblasts centriolar cartwheels are duplicated in early interphase, forming a new procentriole. This procentriole subsequently converts into a mature centriole during mitosis through progressive loading of Asl. Thus, by the end of telophase, both neuroblast centrosomes contain 2 replication-competent mature centrioles, an older mother and younger daughter centriole, which separate in the following interphase starting the cycle again.

### Asymmetric Cnb localization is established in early mitosis through dynamic exclusion from the mother centriole and enrichment on the daughter centriole

Molecular and functional centrosome asymmetry is detectable in interphase neuroblasts but when and how this asymmetry is established is unclear (S1B Fig). To this end, we analyzed the localization of Cnb fused to yellow fluorescent protein (YFP::Cnb) [8] with 3D-SIM throughout mitosis. As expected, YFP::Cnb was localized with Asl on the active, apical centrosome in interphase neuroblasts but absent on the basal interphase centrosome (Fig 1A–1D). To our surprise, we also found apical—but never basal—prophase and prometaphase centrosomes where Cnb was localized on both centrioles (green arrowheads and bars in Fig 1B, 1G and S1 Data). However, Cnb was predominantly localized on one centriole only from metaphase onward (brown arrowheads and bars in Fig 1B, 1G and S1 Data). On the basal centrosome, Cnb appeared in prophase and was consistently localized to a single centriole in all subsequent mitotic stages (Fig 1D, 1G and S1 Data).

Because Asl sequentially loads onto the forming daughter centriole [28,33], we tested whether Asl can be used as an independent marker for centriolar age. To this end, we

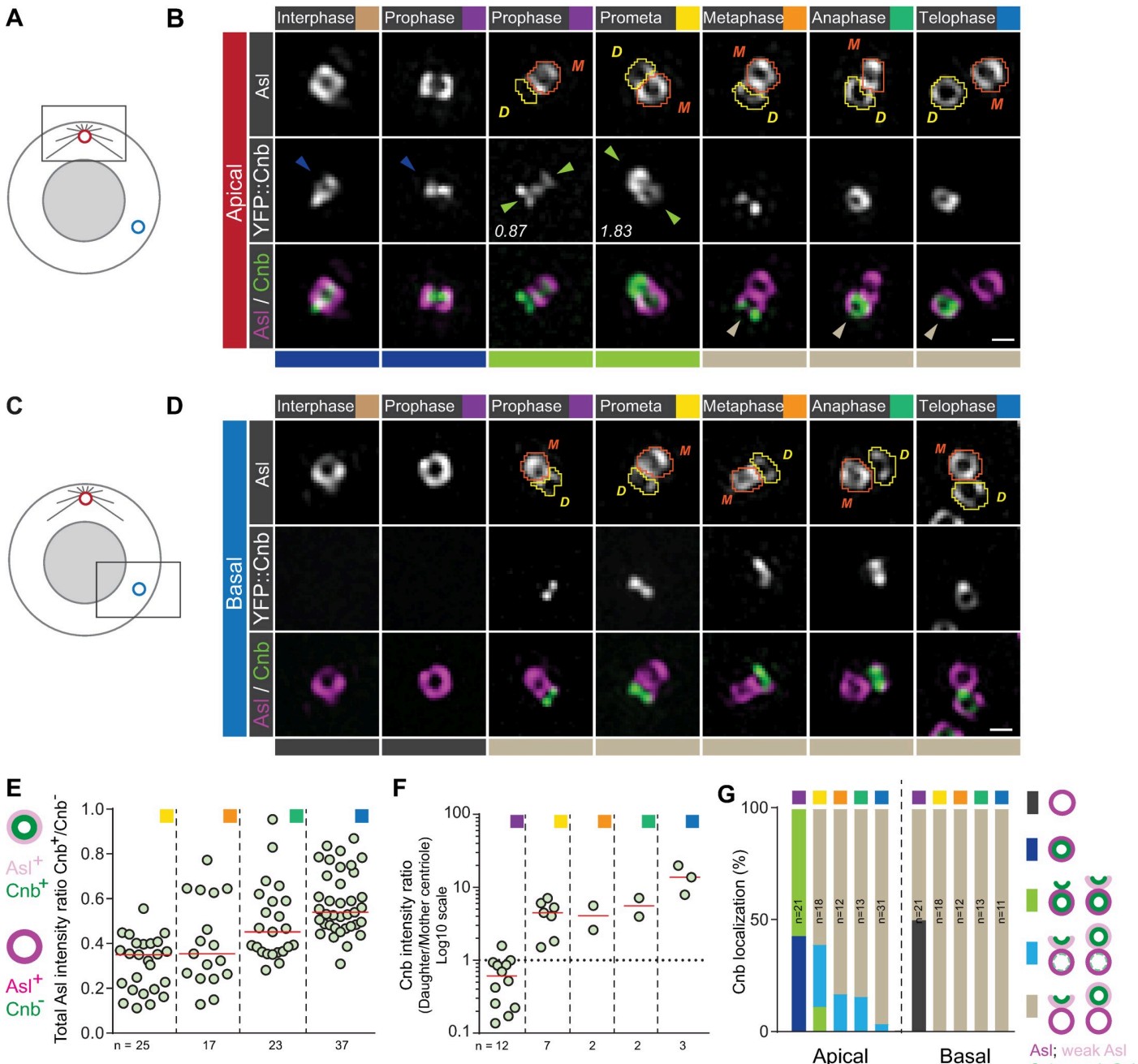

**Fig 1. Cnb localizes on the daughter centriole in early mitosis.** How centriole duplication and molecular asymmetry are coupled is unclear for both the apical (A) and basal (C) centrosome. Representative 3D-SIM images of apical (B) and basal (D) third instar neuroblast centrosomes, expressing YFP::Cnb (middle row; white. Green in merge) and stained for Asl (top row; white. Magenta in merge). Orange and yellow shapes highlight mother and daughter centrioles, respectively, and were used to measure signal intensities. The numbers indicate the total Cnb asymmetry ratios (daughter/mother). Colored arrowheads and bars underneath the images highlight the different stages shown in panel G. (E) For prometaphase to telophase centrosomes (apical and basal combined), containing a single Cnb$^+$ centriole, total Asl intensity of the Cnb$^+$ (presumably the daughter) centriole was divided by the total Asl intensity of the Cnb$^-$ (presumably the mother) centriole. Medians are shown with a red horizontal line. (F) Scatter plot showing total Cnb intensity of the daughter centriole (less Asl), divided by total Cnb intensity on the mother centriole (more Asl). Only apical centrioles containing Cnb on both centrioles were measured. (G) Graph showing the timeline of Cnb's localization dynamics on the apical and basal centrosome: the bars show the percentage of neuroblasts containing an apical centrosome containing one centriole Cnb$^+$ (dark blue), a basal centrosome containing one centriole without Cnb (dark gray), a centrosome with Cnb on both centrioles (transition stage with a daughter/mother ratio < 2; light green), predominant Cnb localization on the daughter centriole (strong asymmetry with a daughter/mother ratio between 2 and 10; light blue), or in which Cnb is only present on the daughter centriole (complete asymmetry with a daughter/mother ratio > 10; light brown) at defined mitotic stages. For this and all subsequent cartoons: closed and open circles represent established mother and forming daughter centrioles, respectively. Cell-cycle stages are indicated with colored boxes. Scale bar is 0.3 μm The data presented here were obtained from 5

independent experiments. Numerical data for panels E, F, and G can be found in the file S1 Data.xlsx. Asl, Asterless; Cnb, Centrobin; YFP, yellow fluorescent protein; 3D-SIM, 3D structured illuminated microscropy.

calculated the Asl intensity ratio between both centrioles (see Methods)—on the apical and basal centrosome—for all mitotic stages where we could find a clear Cnb asymmetry (Asl intensity ratio of $Cnb^+/Cnb^-$ from prometaphase until telophase). These calculations revealed a clear Asl intensity asymmetry with the $Cnb^+$ centriole always containing less Asl and the $Cnb^-$ containing more Asl (Fig 1E and S1 Data).

Using the Asl intensity ratio as a method to distinguish between mother and daughter centrioles, we next correlated Cnb localization with centriolar age at all mitotic stages. We found that in prophase, when Cnb was detectable on both centrioles, Cnb was predominantly associated with the centriole containing more Asl (the mother centriole). However, during prometaphase, more Cnb was localized on the centriole containing less Asl (the daughter centriole). Cnb was sometimes visible before Asl was robustly recruited to the daughter centriole (green arrowheads in third column of Fig 1B). From metaphase until mitosis exit, Cnb was strongly enriched or exclusively present on the daughter centriole (brown bars and arrowheads Fig 1B, 1D, 1F and 1G and S1 Data).

From these data, we conclude that neuroblast centrosomes generate 2 molecularly distinct centrioles during early mitosis. The dynamics generating this centriole asymmetry differ between the apical and basal centrosomes: On the apical centrosome, Cnb is initially only present on the mother centriole before appearing on the daughter and disappearing on the mother centriole. In contrast, Cnb directly appears on the daughter centriole of the basal centrosome. This establishment of molecular centriole asymmetry occurs during the centriole-to-centrosome conversion period.

### A fraction of daughter-centriole–associated Cnb potentially originates from the mother centriole

We next investigated Cnb's localization dynamics, considering the following hypotheses: (1) Cnb associated with the mother centriole could directly or indirectly translocate to the newly forming daughter centriole during mitosis. (2) Alternatively, mother-centriole–associated Cnb could be down-regulated, while the daughter centriole recruits new Cnb from a pool of cytoplasmic Cnb that has not been associated with the mother Cnb. (3) Cnb's dynamic localization could be due to a combination of these 2 models (Fig 2A). To distinguish between these scenarios, we first performed live-cell imaging of endogenously tagged Cnb::EGFP (see Methods) in conjunction with the mitotic spindle marker mCherry::Jupiter [16]. We found that in late interphase, shortly before the neuroblast enters mitosis, Cnb was strongly localized on the apical neuroblast centrosome. At this cell-cycle stage, the apical centrosome only consists of a single $Asl^+$, $Cnb^+$ mother centriole (S1C Fig and Fig 1B). As neuroblasts entered mitosis, Cnb was down-regulated on the apical centrosome with the lowest intensity occurring between prometaphase and anaphase. Cnb intensity then increased again from anaphase onward (Fig 2B, 2C and 2F and S1 Data).

To test whether daughter-centriole Cnb originates from the mother centriole or is recruited from other sources, we performed fluorescence recovery after photobleaching (FRAP) experiments. We reasoned that if Cnb is recruited to the daughter centriole from a source other than the mother centriole, then bleaching mother-centriole–associated Cnb::EGFP in early prophase—corresponding to the time point when no Cnb is yet associated with the forming daughter centriole (see Fig 1)—should result in EGFP emission recovery during early mitosis.

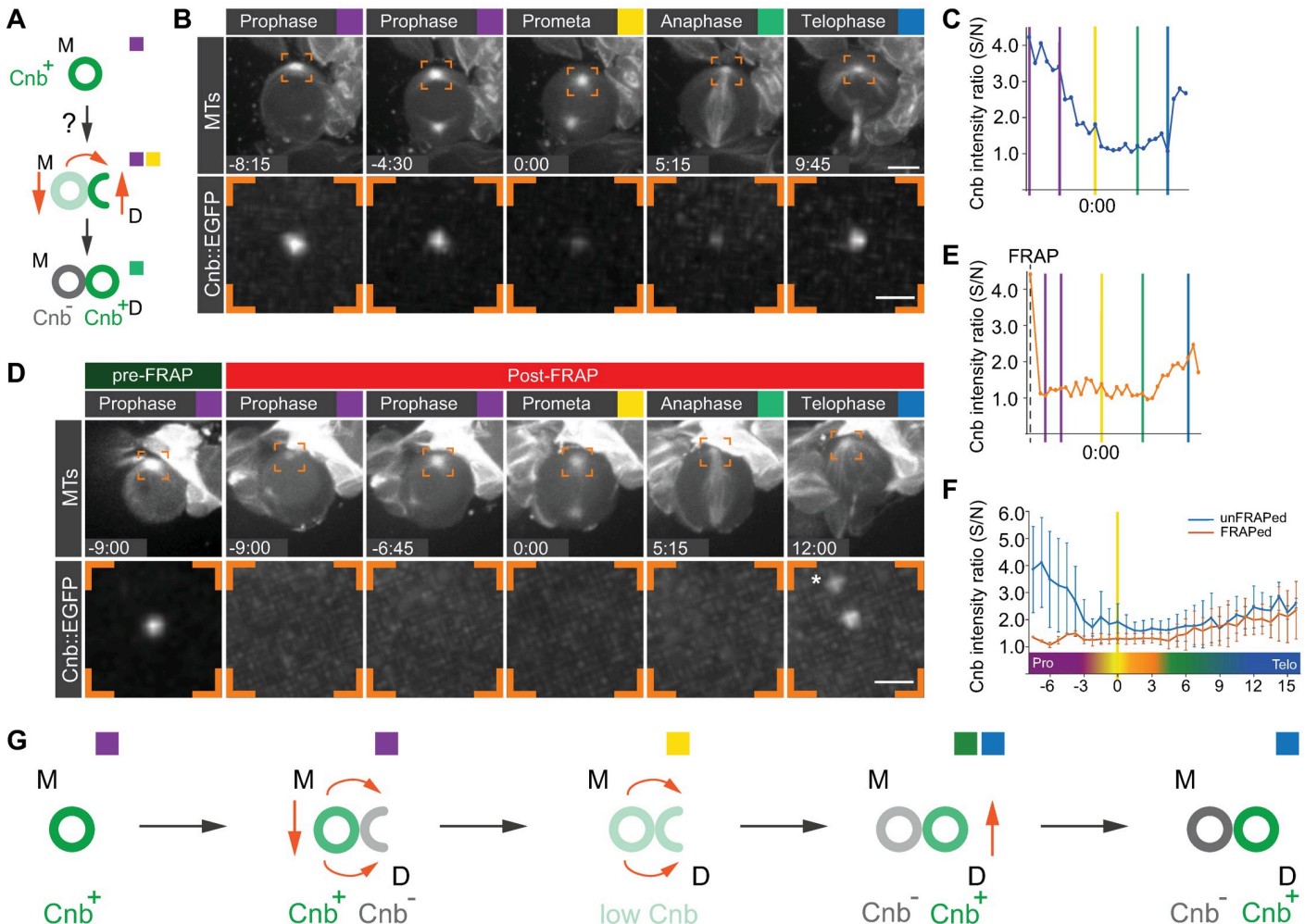

**Fig 2. On the apical centrosome, Cnb localized on the daughter centriole partially originates from the mother centriole.** (A) Dynamic changes in Cnb localization on the mother and daughter centrioles could be either due to a direct transfer mechanism (orange curved arrow) or through up- and down-regulation (vertical orange arrows). Representative unFRAPed (B) and FRAPed (D) wild-type neuroblast expressing endogenously tagged Cnb::EGFP (white; bottom row) together with the MT marker mCherry::Jupiter (white; top row). The orange brackets highlight the apical centrosome where Cnb::EGFP (generated by CRISPR/Cas9; bottom row) is measured. The asterisk refers to an unspecific Cnb::EGFP aggregate. Intensity profile of the unFRAPed (C) and FRAPed (E) apical Cnb::EGFP signal of the neuroblasts shown in (B) and (D). Colored vertical bars indicate specific cell-cycle stages. The vertical dashed line refers to the time point when bleaching was performed. (F) Mean intensity plot of 11 unFRAPed (blue) and FRAPped (orange) apical centrosomes. Cnb intensity ratio prior FRAPing was not plotted here. Error bars indicate standard deviation of the mean. Cnb intensity was plotted as a ratio of centrosomal/cytoplasmic Cnb (S/N; see Methods) in panels C, E, and F. (G) Graphical model for Cnb dynamics on the apical centrosome: Cnb levels decrease during prophase. Starting in prophase already, a pool of mother-centriole–associated Cnb transfers from the mother to the daughter centriole until anaphase. From anaphase onward, Cnb levels increase again likely through recruitment of new Cnb. Time scale is mm:ss. Scale bars in (B) and (D) are 5 μm (top row) and 1 μm (bottom row). The data presented here were obtained from 3 independent experiments. Numerical data for panels C, E, and F can be found in the file S1 Data.xlsx. Cnb, Centrobin; EGFP, enhanced green fluorescent protein; FRAP, fluorescence recovery after photobleaching; MT, microtubule; S/N, Signal/Noise.

Alternatively, if Cnb associated with the mother centriole directly or indirectly transfers to the daughter centriole, then bleaching mother-centriole–associated Cnb::EGFP in early prophase should result in no recovery of EGFP emission until at least metaphase (when Cnb is only present on the daughter centriole). We observed that bleaching of Cnb on the apical centrosome in late interphase or early prophase extinguished Cnb fluorescence, which only recovered from anaphase onward (Fig 2D–2F and S1 Data). In contrast, bleaching of apical Cnb in telophase —when Cnb intensity is on the rise again (Fig 2B and 2C and S1 Data)—resulted in faster

emission recovery (S2A–S2D Fig and S1 Data). Cnb recovery during telophase resembled mitotic Asl recovery (bleached in prophase); although Asl levels fluctuate during mitosis, they are consistently high and recover quickly and steadily when bleached in early mitosis (S2E–S2K Fig and S1 Data). We also tagged Cnb endogenously with mDendra2 (see also below), but the signal was too low to perform photoconversion experiments. Regardless, the lack of Cnb fluorescence recovery in early mitosis suggests that very little to no new Cnb is recruited to the apical centrosome prior to anaphase. However, Cnb's steady recovery during telophase suggests that new Cnb is recruited at the end of mitosis. Taken together, these observations are inconsistent with a model proposing that daughter-centriole–bound Cnb is exclusively recruited from a source other than the mother centriole—especially in early mitosis. Although direct proof is lacking, the data are more compatible with the idea that Cnb detectable on the daughter centriole in prophase and prometaphase (see Fig 1B) originates from the mother centriole. Furthermore, the increase in Cnb intensity from anaphase onward and the fast recovery after photobleaching in telophase suggests that new Cnb is recruited from a Cnb protein pool other than the apical mother centriole later in mitosis. In conclusion, we think that our data are most consistent with a hybrid model and propose that a small fraction of Cnb transfers from the mother to the daughter centriole in early mitosis through an unknown mechanism. From anaphase onward, the daughter centriole recruits additional Cnb from sources other than the mother centriole (Fig 2G).

## Polo-dependent phosphorylation of Cnb is necessary for its timely mother centriole depletion and daughter-centriole enrichment

Previously, it was shown that Cnb is a substrate of Polo [19]. We thus tested whether Cnb's dynamic localization depends on Polo phosphorylation. To this end, we analyzed the localization of YFP::Cnb$^{T4A,T9A,S82A}$, a mutant version of Cnb in which all 3 consensus phosphorylation sites for Polo were substituted by alanine, in *cnb* mutant neuroblasts. In *cnb* mutants, both neuroblast centrosomes lose MTOC activity in interphase [19]. Because we cannot accurately distinguish between apical and basal centrosomes in *cnb* mutants expressing YFP::Cnb$^{T4A,T9A,S82A}$, we will define centrosome 1 as the centrosome with a stronger localization defect (stronger mother centriole retention of Cnb), whereas centrosome 2 has a less pronounced localization phenotype. In contrast to wild-type cells, phosphomutant Cnb was detectable on both centrosomes in early prophase neuroblasts (68.4% of the cases, sum of dark blue and light green for centrosome 2, Fig 3B and S1 Data). In most wild-type neuroblasts, Cnb was depleted on the mother and enriched on the daughter centriole by metaphase but in *cnb* mutant neuroblasts expressing YFP::Cnb$^{T4A,T9A,S82A}$; 71.4% ($n = 7$) of analyzed neuroblasts show incomplete mother-centriole Cnb depletion and daughter-centriole Cnb enrichment on at least one centrosome by telophase (Fig 3A and 3B; light blue, centrosome 1 and S1 Data). A similar albeit weaker YFP::Cnb localization phenotype was also observed in hypomorphic *polo* mutant neuroblasts (*polo$^{16-1}$/polo$^1$*; S3A–S3D Fig and S1 Data).

To more directly visualize the localization dynamics of YFP::Cnb$^{T4A,T9A,S82A}$, we imaged *cnb* mutants expressing YFP::Cnb$^{T4A,T9A,S82A}$ live. Because we could not obtain reliable live 3D-SIM data, we reasoned that we could visualize 2 Cnb$^+$ centrioles when the mother and daughter centrioles separate from each other at the end of telophase with conventional spinning disc microscopy (Fig 3C–3F). Indeed, in contrast to the majority of wild-type neuroblasts, retaining a single Cnb$^+$ centriole on the apical cortex (75%; $n = 20$), we found 2 separating Cnb$^+$ centrioles in most *cnb* mutants expressing YFP::Cnb$^{T4A,T9A,S82A}$ (73%; $n = 22$; Fig 3D, 3F and 3G and S1 Data). These data suggest that altered localization dynamics of phosphomutant Cnb during mitosis gives rise to 2 Cnb$^+$ centrioles by the end of telophase. Thus, incomplete

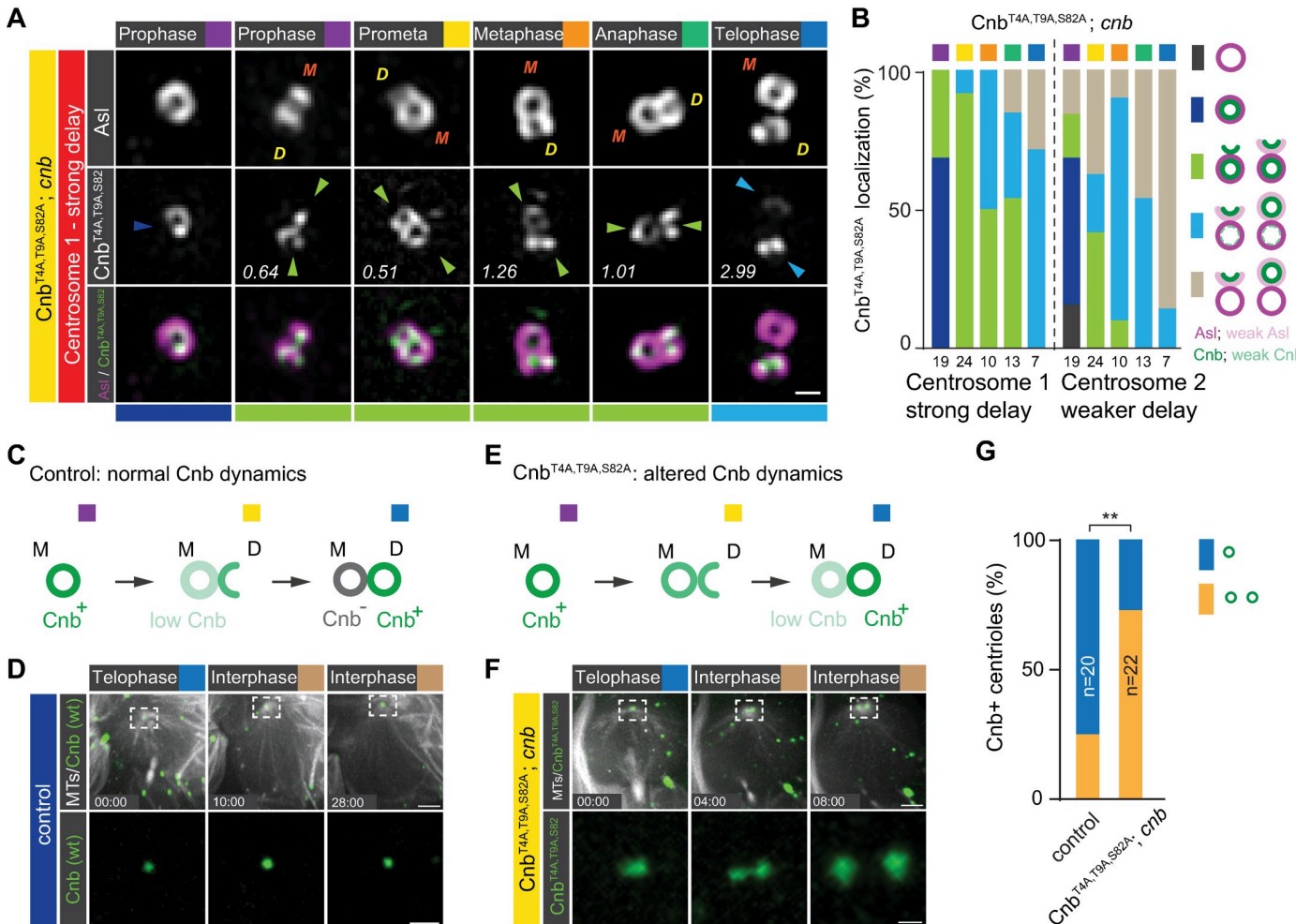

**Fig 3. Cnb's asymmetric localization in favor of the daughter centriole is controlled by Polo-dependent phosphorylation.** (A) Representative 3D-SIM images of centrosome 1 of third instar *cnb* mutant larval neuroblasts, expressing YFP::Cnb^T4A,T9A,S82A (white; middle row, green; bottom row). Brains were stained for Asl (top row: white; bottom row: magenta). Orange "M" and yellow "D" stand for mother and daughter centriole, respectively. The numbers indicate the daughter/mother intensity ration of the representative image. Colored arrowheads and bars underneath the images highlight the degree of Cnb localization (see panel B). (B) Quantification of YFP::Cnb^T4A,T9A,S82A localization defects in *cnb* mutant neuroblasts. The bars show the percentage of neuroblasts containing a single Cnb+ centriole (dark blue), a single centriole without Cnb (dark gray), Cnb on both centrioles (transition stage with a daughter/mother ratio < 2; light green), predominant Cnb localization on the daughter centriole (strong asymmetry with a daughter/mother ratio between 2 and 10; light blue) or in which Cnb is completely shifted to the daughter centriole (complete asymmetry with a daughter/mother ratio > 10; light brown). (C) Normal Cnb localization dynamics should result in one Cnb+ and one Cnb− centriole, whereas (E) delayed Cnb localization dynamics should result in 2 Cnb+ centrioles in early interphase when mother and daughter centrioles separate from each other. Representative live-cell imaging sequence of a (D) control neuroblast, expressing wild-type YFP::Cnb (green) and (F) *cnb* mutant neuroblast expressing YFP::Cnb^T4A,T9A,S82A (green). Both samples also co-express the spindle marker UAS-mCherry::Jupiter (white) to visualize microtubules. (G) Quantification of centriole splitting phenotype; blue bars represent neuroblasts retaining a single Cnb+ centriole on the apical centrosome. Orange bars represent neuroblasts generating 2 Cnb+ centrioles in early interphase. Live-cell imaging experiments were repeated 4 times independently. Fisher's exact test: *p* = 0.0048. Cell-cycle stages are indicated with colored boxes. Time scale is mm:ss. Scale bar is 0.3 μm in panel A and 5 μm (top row) or 2 μm (bottom row) in panels D and F. Numerical data for panels B and G can be found in the file S1 Data.xlsx. Asl, Asterless; Cnb, Centrobin; YFP, yellow fluorescent protein; 3D-SIM, 3D structured illuminated microscropy.

mother-centriole depletion and daughter-centriole enrichment of Cnb will result in neuroblasts reentering the next mitosis with Cnb on both centrosomes. Taken together, we conclude that Polo-dependent phosphorylation of Cnb is necessary for the establishment of molecularly distinct centrioles during mitosis, impacting subsequent molecular interphase asymmetry.

## Polo becomes enriched on the daughter centriole whereas Plp remains localized on the mother centriole

Having implicated Polo in Cnb's localization dynamics, we then analyzed the localization of Polo (Polo::GFP) and Plp (Plp::EGFP; generated by CRISPR/Cas9; see Methods). The latter has previously been shown to be involved in centrosome asymmetry establishment [22]. Both Polo and Plp were GFP-tagged at the endogenous locus (the work by Buszczak and colleagues [34] and Methods). In early prophase neuroblasts, Polo was localized on the existing centriole on both centrosomes (Fig 4A and 4B and the work by Ramdas Nair and colleagues [20]). Subsequently, Polo intensity increased on the forming daughter centriole and its asymmetric localization peaked in metaphase and anaphase. Interestingly, the apical centrosome showed a less pronounced asymmetric distribution in prometaphase compared to the basal centrosome, which could reflect differences in the localization mechanism (Fig 4A–4C and S1 Data).

In contrast to Polo and Cnb, Plp predominantly remained localized on the mother centriole on both centrosomes, although it increased also on the daughter centriole in late mitosis (S3E–S3G Fig and S1 Data). Co-imaging Polo together with Plp, and Cnb with Plp showed that Polo

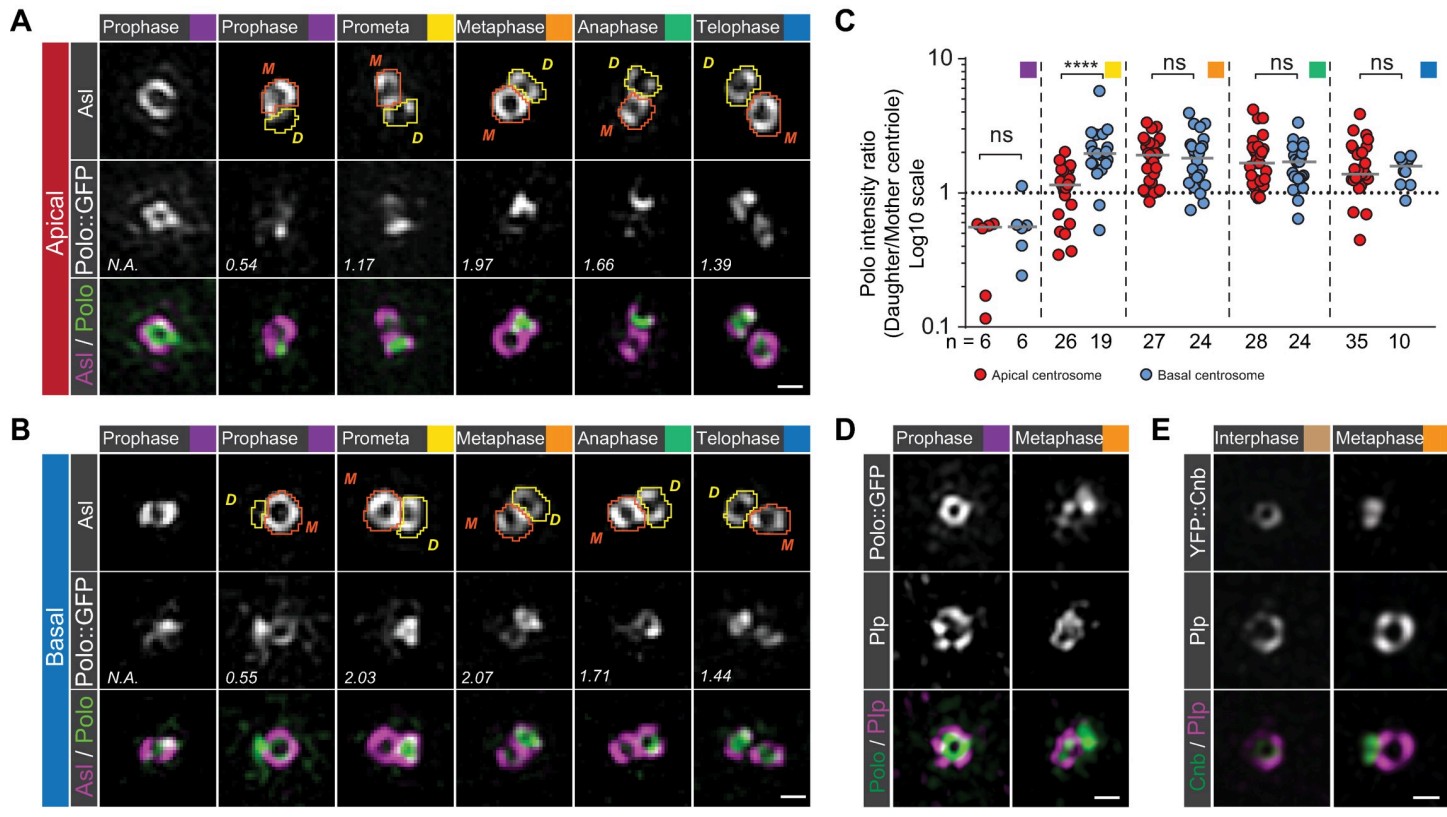

**Fig 4. Polo and Cnb separate from Plp in mitosis.** Representative 3D-SIM images of (A) apical or (B) basal third instar larval neuroblast centrioles, expressing Polo::GFP (protein trap line [34]; middle row; green in merge). Centriole contours were drawn based on Asl signal (orange and yellow lines for mother and daughter centriole, respectively) and used to measure Polo::GFP and Asl intensities. The numbers represent total Polo intensity ratios (daughter/mother centriole) in the shown image. Polo asymmetry ratios for the apical (red dots) and the basal (blue dots) centrosome are plotted in panel C from 3 independent experiments. Medians are shown with a gray horizontal line. Mann–Whitney test: Prophase: apical versus basal; $p = 0.6991$. Prometaphase: apical versus basal; $p = 5.688 \times 10^{-6}$. Metaphase: apical versus basal; $p = 0.9329$. Anaphase: apical versus basal; $p = 0.8628$. Telophase: apical versus basal; $p = 0.8614$. Representative interpolated images of apical interphase or early prophase and late metaphase or early anaphase centrosomes, expressing (D) Polo::GFP (protein trap line [34]; green in merge) or (E) YFP::Cnb (green in merge) and stained for Plp (magenta in merge). These experiments were performed 3 times independently for Polo::GFP and once for YFP::Cnb. Cell-cycle stages are indicated with colored boxes. Scale bar is 0.3 μm. Numerical data for panel C can be found in the file S1 Data.xlsx. Asl, Asterless; Cnb, Centrobin; GFP, green fluorescent protein; Plp, PCNT-like protein; YFP, yellow fluorescent protein; 3D-SIM, 3D structured illuminated microscopy.

and Cnb separated from Plp in metaphase and anaphase (Fig 4D and 4E). These data suggest that similar to Cnb on the apical centrosome, Polo is changing its localization from the mother to the daughter centriole during mitosis. However, in contrast to Cnb, Polo's localization dynamics appear similar on both centrosomes. Plp remains enriched on the mother centriole on both the basal and apical centrosome.

## Polo's localization on the daughter centriole depends on Wdr62 and Cnb, with Polo and Cnb co-depending on each other

We next asked how asymmetric Polo localization establishment is regulated. To this end, we analyzed Polo localization in neuroblasts depleted for Cnb by RNA interference (*cnb* RNAi) and Wdr62 (*wdr62* mutants). Wdr62 is implicated in primary microcephaly [35,36], and both Cnb and Wdr62 are necessary for MTOC asymmetry by regulating Polo's and Plp's centrosomal localization in interphase neuroblasts [19,20]. Lack of Cnb or Wdr62 did not compromise the gradual loading of Asl onto the newly formed centriole in mitotic neuroblasts and Plp localization was still highly asymmetric in favor of the mother centriole. However, the asymmetric centriolar localization of Polo, especially from prometaphase to anaphase neuroblasts, was significantly perturbed in the absence of Cnb and Wdr62 (Fig 5A–5C and S1 Data). Lack of Cnb—but not Wdr62—also compromised Polo's asymmetric localization in telophase, suggesting a preferential requirement for Wdr62 in metaphase and anaphase.

Our *polo* mutant, Cnb phosphomutant, and Cnb RNAi data are consistent with previous studies, indicating a co-dependency of Polo and Cnb [19,22]. To test whether Cnb mislocalization is sufficient to prevent Polo's enrichment on the daughter centriole, we expressed mCherry::Cnb::PACT (see Methods) together with Polo::EGFP (tagged endogenously, using CRISPR/Cas9 technology; see Methods). Because our 3D-SIM data showed Plp to be predominantly associated with the mother centriole, we reasoned that tethering Cnb to the mother centriole with Plp's PACT domain [37] would compromise the establishment of a Cnb⁻ mother and Cnb⁺ daughter centriole. We speculated that Cnb's localization would remain enriched on the mother centriole or at least become near symmetrically localized. Indeed, our 3D-SIM experiments revealed that by tethering the PACT domain to Cnb (mCherry::Cnb::PACT or YFP::Cnb::PACT [19]) prevented the establishment of normal asymmetric Cnb localization (Fig 5D & S4A and S4B Fig and S1 Data). Neuroblasts expressing mCherry::Cnb::PACT also failed to establish high daughter/mother centriole Polo asymmetry. Polo::EGFP was either localized symmetrically (with equal amounts on both the mother or daughter centriole) or, as observed in most cases, inverted asymmetrically (with higher Polo::EGFP amounts on the mother centriole; Fig 5D–5E and S1 Data). Taken together, loss or mislocalization of Cnb and depletion of *wdr62* significantly increased the number of centrosomes with inverted Polo asymmetry ratios (wild-type controls: 8.6% and 2%, respectively; *cnb* RNAi: 40%; *wdr62*: 31.5%; Cnb::PACT: 97.8%; Fig 5F and 5G and S1 Data). We conclude that Wdr62 and Cnb are necessary to establish "low-Polo" mother and "high-Polo" daughter centrioles. Furthermore, Polo and Cnb both co-depend on each other to correctly establish this centriolar asymmetry.

## Disrupting centriolar asymmetry impacts biased MTOC activity in interphase and spindle orientation in metaphase

Next, we set out to investigate the significance of centriole asymmetry establishment by preventing the enrichment of Cnb and Polo on the daughter centriole using the PACT domain (see above). It was previously shown that expression of YFP::Cnb::PACT in neuroblasts converted the inactive mother interphase centrosome into an active MTOC, resulting in the presence of 2 active interphase MTOCs. This phenotype was entirely attributed to Cnb's function

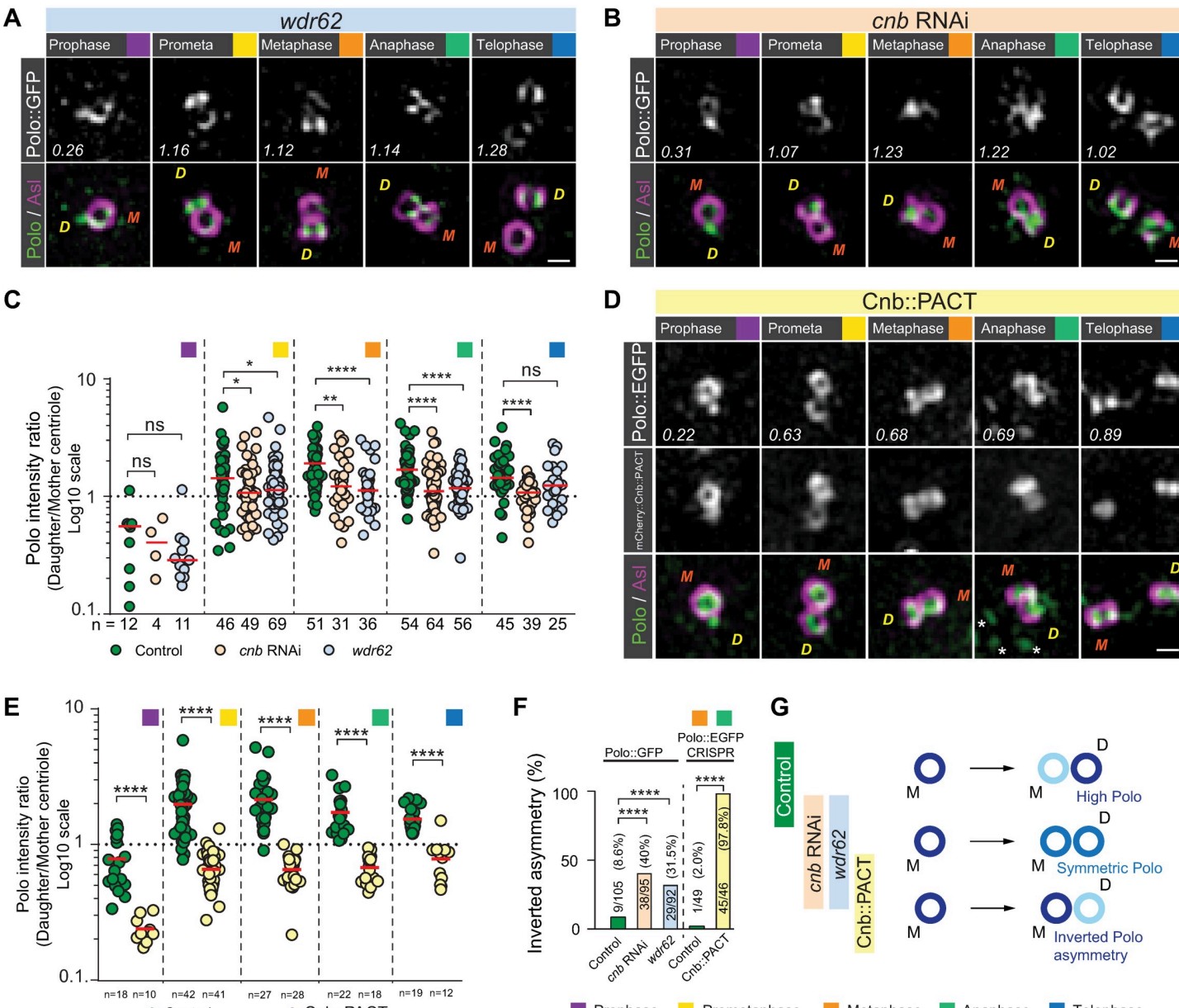

**Fig 5. Polo's enrichment on the daughter centriole during mitosis depends on Cnb and Wdr62.** Representative 3D-SIM images of third instar larval neuroblasts mutant for (A) *wdr62* or (B) expressing RNAi against Cnb (*cnb* RNAi). In both conditions, Polo::GFP (protein trap line [34]; green in merge) expressing neuroblasts were stained for Asl (magenta in merge). For panels A, B, and D, orange "M" and yellow "D" represent mother and daughter centriole, respectively. Polo intensity ratios (daughter/mother centriole) are shown in the representative images and plotted in panel C for control (wild-type background; green dots), *cnb* RNAi (beige dots), and *wdr62* mutants (blue dots). Because apical and basal centrosomes could not be distinguished in *cnb* RNAi and *wdr62* mutants, measurements from these conditions were compared to the pooled (apical and basal) control Polo measurements (replotted from Fig 4C). These experiments were performed 3 times independently for wild-type control and *cnb* RNAi, and 6 times for *wdr62*. Medians are shown in red. Mann–Whitney test: Prophase: wild-type control versus *cnb* RNAi: $p = 0.6835$; wild-type control versus *wdr62*: $p = 0.1179$. Prometaphase: wild-type control versus *cnb* RNAi: $p = 0.0318$; wild-type control versus *wdr62*: $p = 0.0439$. Metaphase: wild-type control versus *cnb* RNAi: $p = 0.0040$; wild-type control versus *wdr62*: $p = 8.496 \times 10^{-5}$. Anaphase: wild-type control versus *cnb* RNAi: $p = 4.19 \times 10^{-6}$; wild-type control versus *wdr62*: $p = 1.79 \times 10^{-6}$. Telophase: wild-type control versus *cnb* RNAi: $p = 1.17 \times 10^{-6}$; wild-type control versus *wdr62*: $p = 0.0524$. (D) Representative 3D-SIM images of third instar larval neuroblast centrosomes, expressing Polo::EGFP generated by CRISPR/Cas9 technology (white, top row; green, bottom row) and mCherry::Cnb::PACT (white in middle row), stained for Asl (magenta in merge; bottom row). Asterisks in the merged anaphase image indicate kinetochores reaching the spindle pole. Polo intensity ratios (daughter/mother centriole) are shown in the representative images and plotted in panel E for control (wild-type background; green dots) and mCherry::Cnb::PACT-expressing neuroblasts (yellow dots). Medians are shown in red. This experiment was performed 2 times independently for wild-type and mCherry::Cnb::PACT-expressing neuroblast in parallel. Mann–Whitney test: Prophase: wild-type control versus mCherry::Cnb::PACT: $p = 1.524 \times 10^{-7}$. Prometaphase: wild-type control versus mCherry::Cnb::PACT: $p < 1.0 \times 10^{-15}$. Metaphase: wild-type control versus mCherry::Cnb::PACT: $p = 2.0 \times 10^{-15}$. Anaphase: wild-type control versus mCherry::Cnb::PACT: $p = 1.764 \times 10^{-11}$. Telophase: wild-type control versus mCherry::Cnb::PACT: $p = 3.854 \times 10^{-6}$. The percentage of metaphase and anaphase centrosomes with

inverted Polo asymmetry (daughter/mother ratio < 1) are plotted in panel F. Fisher's exact test: Polo::GFP [34] control versus *cnb* RNAi: $p = 1.272 \times 10^{-7}$. Polo::GFP [34] control versus *wdr62*: $p = 5.228 \times 10^{-5}$. Polo::EGFP (CRISPR/Cas9, this study) control versus mCherry::Cnb::PACT: $p = 7.307 \times 10^{-25}$. (G) Summary of phenotypes; efficient enrichment of Polo on the daughter "D" centriole is prevented in neuroblasts devoid of Wdr62 or Cnb, or with mislocalized Cnb. Cell-cycle stages are indicated with colored boxes. Scale bar is 0.3 μm. Numerical data for panels C, E, and F can be found in the file S1 Data.xlsx. Asl, Asterless; Cnb, Centrobin; EGFP, enhanced green fluorescent protein; GFP, green fluorescent protein; PACT, Pericentrin-AKAP-450 containing targeting; RNAi, RNA interference; Wdr62, WD40 repeat protein 62; 3D-SIM, 3D structured illuminated microscopy.

in interphase [19] (S4C Fig, S1 & S2 Movies). Here, we hypothesized that fusing Cnb with the PACT domain affects the correct establishment of molecular centrosome asymmetry during mitosis, manifested in symmetric MTOC activity in the subsequent interphase. To test this hypothesis, we developed a nanobody trapping experiment, using the anti-GFP single domain antibody fragment (vhhGFP4) [38,39] fused to the PACT domain of Plp [37] to predominantly trap GFP- or YFP-tagged proteins on the mother centriole (S4D–S4F Fig). Expressing PACT:: vhhGFP4 in neuroblasts together with YFP::Cnb mimics the YFP::Cnb::PACT phenotype; almost 93% (*n* = 69) of neuroblasts expressing PACT::vhhGFP4 together with YFP::Cnb showed 2 active interphase MTOCs (YFP::Cnb expression only: no MTOC gain of function observed; *n* = 16; S4G and S4H Fig; S3 Movie and S1 Data). Conversely, trapping Asl::GFP with PACT::vhhGFP4 on the mother centriole did not cause a MTOC phenotype in 83% of neuroblasts (*n* = 104; S4I and S4J Fig; S4 Movie and S1 Data).

Having validated the nanobody tool, we next co-expressed a GFP-tagged version of Polo, either a published GFP::Polo *trans*-gene [40] or our endogenously tagged CRISPR Polo::EGFP line, with PACT::vhhGFP4. The 3D-SIM data revealed that, under these experimental conditions, Polo::EGFP was strongly localized to the mother centriole in prophase. Subsequently, Polo::EGFP was symmetrically localized between mother and daughter centriole from prometaphase onwards (Fig 6A and 6B and S1 Data). Nanobody-mediated trapping of Polo on the mother centriole also induced the formation of 2 active interphase MTOCs (GFP::Polo *trans*-gene: 84%; *n* = 31. Polo::EGFP CRISPR line: 72%; *n* = 82; S4K and S4L Fig, Fig 6C–6E & S5–S7 Movie and S1 Data). Although cell-cycle progression was not affected in these neuroblasts, we measured a significant misorientation of the mitotic spindle in early metaphase ($p = 1.1125 \times 10^{-8}$; *n* = 26 for control, *n* = 37 for vhhGFP; Fig 6G, 6G and 6I and S1 Data). However, similar to *bld10* mutant neuroblasts, displaying 2 active interphase MTOCs also [21], mitotic spindles realigned along the apical-basal polarity axis, ensuring normal asymmetric cell divisions along a conserved axis between successive mitoses (Fig 6H and 6J and S1 Data). The 3D-SIM imaging also revealed that in Polo::EGFP and vhhGFP4::PACT-expressing neuroblast, both interphase centrosomes (now containing one centriole each) contain high levels of centriolar and diffuse PCM Polo, consistent with our recent observation for the apical interphase wild-type centrosome [20] (Fig 6K and 6L). Based on these experiments, we conclude that preventing the normal establishment of Cnb and Polo asymmetry using the PACT domain perturbs biased MTOC activity in interphase.

## Optogenetically induced Polo and Cnb trapping during mitosis affects MTOC activity in the subsequent interphase

Based on these nanobody results, we reasoned that trapping Polo and Cnb on the mother centriole at defined cell-cycle stages should allow us to test more specifically whether the establishment of Polo and Cnb asymmetry during mitosis has an impact on MTOC activity in the subsequent interphase. To test this hypothesis, we implemented the optogenetic system improved light-induced dimer (iLID) [41] by generating transgenic flies containing the iLID cassette (containing *Avena Sativa's* LOV domain for Light-Oxygen-Voltage) fused with the PACT domain (*UAS-iLID::PACT::HA; UAS-iLID::PACT::GFP*; see Methods). iLID (or SsrA

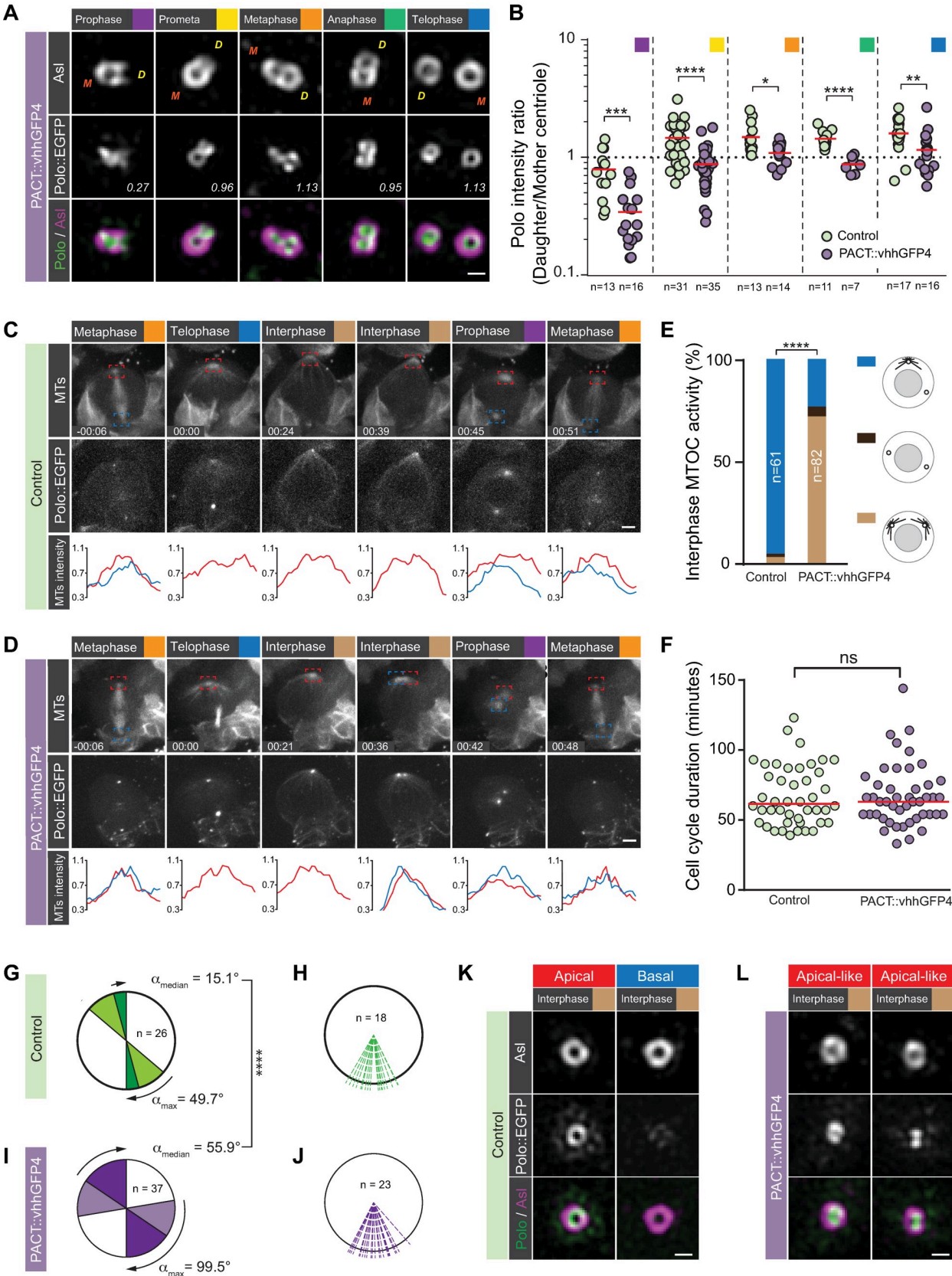

**Fig 6. Establishment of centriolar asymmetry is required for biased interphase MTOC activity and centrosome positioning.** (A) Representative 3D-SIM images of third instar larval neuroblast centrosomes, expressing Polo::EGFP (generated by CRISPR/Cas9) and the nanobody construct PACT::vhhGFP4. Polo::EGFP (middle: white; merge: green) expressing neuroblasts were stained for Asl (white; top row, magenta in the merge). Polo intensity ratios (daughter/mother centriole) are plotted in panel B for control (green dots) and PACT::vhhGFP4 (purple dots). These experiments were performed 2 times independently in parallel for both genotypes. Medians are shown in red. Mann–Whitney test: Prophase: control versus PACT::vhhGFP4: $p = 3.11 \times 10^{-4}$. Prometaphase: control versus PACT::vhhGFP4: $p = 3.49 \times 10^{-6}$. Metaphase: control versus PACT::vhhGFP4: $p = 0.0222$. Anaphase: control versus PACT::vhhGFP4: $p = 6.28 \times 10^{-5}$. Telophase: control versus PACT:: vhhGFP4: $p = 0.0077$. (C) Representative live-cell imaging time series of a dividing control (Polo::EGFP, worGal4, UAS-mCherry::Jupiter) and (D) PACT::vhhGFP4 expressing (Polo::EGFP, worGal4, UAS-mCherry::Jupiter, and PACT::vhhGFP4) neuroblast. The microtubule marker (MTs, first row) and Polo::EGFP (generated by CRISPR/Cas9; second row) are shown for 2 consecutive mitoses. Microtubule intensity of the apical (red line and square) and basal (blue line and square) MTOC are plotted below. "00:00" corresponds to the telophase of the first division. (E) Bar graph showing the quantification of the MTOC phenotype in interphase (blue; wild-type–like asymmetry, dark brown; loss of MTOC activity, light brown; gain of MTOC activity). Fisher's exact test: Polo::EGFP with and without PACT::VhhGFP4 expression; $p = 1.786 \times 10^{-19}$. Cell-cycle length is shown in panel F. The cell-cycle length in PACT::vhhGFP4 (purple dots) is not significantly different from the control (green dots): $p = 0.9727$ (Mann–Whitney test). Medians are shown in red. Panels G and I represent the spindle rotation between NEBD and anaphase. Medians are displayed in dark colors (green; control. Purple; vhhGFP4 expressing neuroblasts); $p = 1.1125 \times 10^{-8}$; $n = 26$ for control; $n = 37$ for vhhGFP. Maximum rotation is shown in light colors. Division orientation between consecutive mitoses shown for control (H) and PACT::vhhGFP4 (J). Panels K and L are representative 3D-SIM images of interphase centrosomes for control and PACT::vhhGFP4 expressing neuroblasts, respectively. The trapping of Polo::EGFP (generated by CRISPR/Cas9) with PACT::vhhGFP4 induces 2 identical apical-like (in respect to MTOC activity and Polo localization) centrosomes with a strong centriolar and PCM signal. The data presented for the live imaging here were obtained from 5 independent experiments. Cell-cycle stages are indicated with colored boxes. Yellow "D" and orange "M" refer to daughter and mother centrioles based on Asl intensity. Time stamps are shown in hh:mm and scale bar is 0.3 μm (A, K, L) and 3 μm (C, D), respectively. Numerical data for panels B, E, F, G, H, I, and J can be found in the file S1 Data.xlsx. Asl, Asterless; EGFP, enhanced green fluorescent protein; MTOC, microtubule organizing center; NEBD, Nuclear envelop break-down; PACT, Pericentrin-AKAP-450 containing targeting; PCM, pericentriolar material; 3D-SIM, 3D structured illuminated microscopy.

for 10SA RNA) binds to the small SspB (for Stringent starvation protein B) domain under blue-light exposure [41]. To test this system in fly neuroblasts, we expressed cytoplasmic SspB:: mCherry together with iLID::PACT::GFP and exposed entire larval brains first to yellow (561 nm) light, followed by simultaneous blue and yellow light (488 and 561 nm) exposure, before switching back to only 561 nm; each exposure period lasted 5 minutes. Blue-light exposure was sufficient to induce the recruitment of cytoplasmic SspB::mCherry to neuroblast centrioles containing iLID::PACT::GFP within 15 seconds. This behavior is strictly blue-light dependent because imaging with 561 nm alone is not sufficient to recruit SspB::mCherry to centrioles and SspB::mCherry relocalized to the cytoplasm within 100 s after blue-light exposure was shut off (S5A Fig).

Next, we generated *SspB::EGFP::Polo* and *SspB::mDendra2::Cnb* flies using CRISPR/Cas9. We reared embryos, expressing iLID::PACT::HA under the control of the neuroblast specific *worGal4* driver together with SspB::EGFP::Polo or SspB::mDendra::Cnb in the dark for 4 days before exposing third instar larval neuroblasts in intact brains to blue light at different cell-cycle stages for 10 to 20 minutes. Subsequently, we monitored MT dynamics using mCherry:: Jupiter for approximately 90 minutes without blue-light exposure. If the dynamic localization of Polo and Cnb during mitosis is important for the correct MTOC establishment in the subsequent interphase (interphase 2), we would expect that light-dependent manipulation of Cnb and Polo localization would mimic the nanobody phenotype, resulting in 2 active MTOCs in interphase 2. Indeed, a significant number of neuroblasts, exposed to blue light from late interphase 1 or prophase 1 onward, showed 2 active MTOCs in the following interphase 2. However, continued blue-light exposure during interphase—early interphase in particular—also disrupted MTOC asymmetry in late interphase just prior to mitotic entry (Fig 7A–7C). Overall, approximately 55% of SspB::EGFP::Polo and iLID::PACT::HA and approximately 46% of SspB::mDendra2::Cnb and iLID::PACT::GFP expressing neuroblasts, exposed to blue light showed an MTOC phenotype ($n = 67$ and $n = 39$, respectively; Fig 7D and 7E, S8 Movie and S1 Data). SspB::EGFP::Polo also displayed a more focused and intense localization when co-expressed with iLID::PACT::HA and exposed to blue light, compared to normal SspB::EGFP::

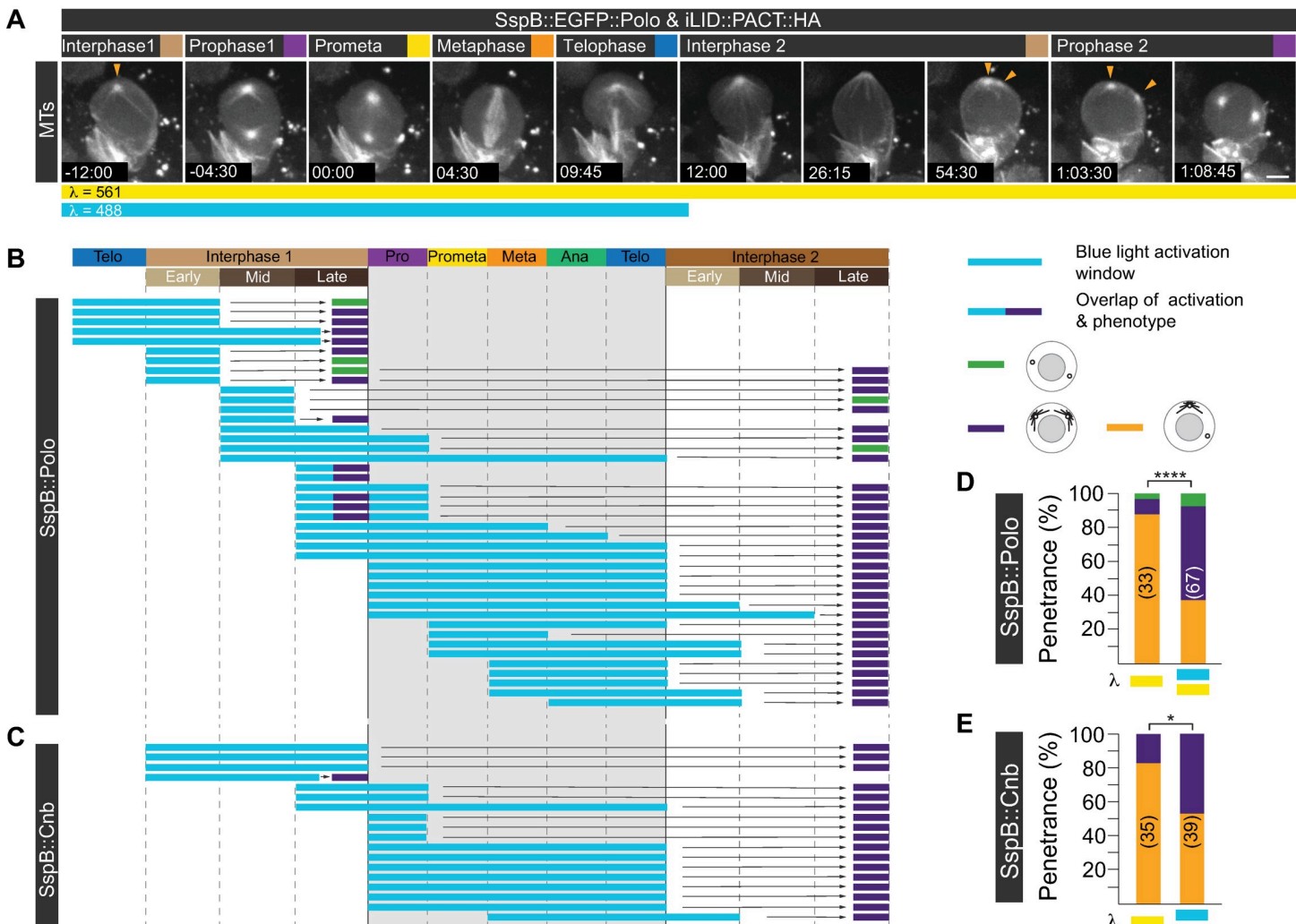

**Fig 7. Establishment of centriolar asymmetry during mitosis is required for biased interphase MTOC activity.** (A) Representative wild-type neuroblast expressing SspB::EGFP::Polo (not shown) together with the microtubule marker mCherry::Jupiter (white) and iLID::PACT::HA (not shown). As indicated with the cyan and yellow color bars underneath the image sequence, this neuroblast was exposed to both 488 nm and 561 nm during the first division but only to 561 nm in the second division. Yellow arrowheads indicate 2 active MTOCs in the interphase 2 and prophase 2. Summary of all optogenetic experiments for (B) SspB::EGFP::Polo and (C) SspB::mDendra2::Cnb and iLID::PACT::HA expressing neuroblasts. Blue-light exposure and resulting phenotype are indicated with the colored bars (see legend on the right). Because neuroblasts do not divide in synchrony, they were exposed to blue light at different cell-cycle stages. This experiment was repeated more than 3 times independently. Bar graphs representing the phenotypic penetrance (in %) of larvae expressing (D) SspB::EGFP::Polo and iLID::PACT::HA or (E) SspB::mDendra2::Cnb and iLID::PACT::GFP with (cyan and yellow bars) or without (yellow bar only) blue-light exposure. The number of scored divisions are indicated on the bars. Fisher's exact test: SspB::Polo with and without blue-light exposure: $p = 1.944 \times 10^{-6}$. SspB::Cnb with and without blue-light exposure; $p = 0.0123$. Time stamps are shown in hh:mm:ss and scale bar is 5 μm. Numerical data for panels D and E can be found in the file S1 Data.xlsx. Cnb, Centrobin; EGFP, enhanced green fluorescent protein; GFP, green fluorescent protein; HA, hemagglutinin; iLID, improved light-induced dimer; MTOC, microtubule organizing center; PACT, Pericentrin-AKAP-450 containing targeting; SspB, Stringent starvation protein B.

Polo localization (S5B Fig). These observed phenotypes are strictly blue-light dependent as SspB::EGFP::Polo or SspB::mDendra2::Cnb expressed in conjunction with iLID::PACT and imaged with 561 nm only, showed predominantly normal MTOC activity (SspB::Polo: 88% normal asymmetry; $n = 33$; SspB::Cnb: 83% normal asymmetry; $n = 35$; Fig 7D and 7E and S1 Data). Taken together, these experiments suggest that perturbing normal Cnb and Polo asymmetry during mitosis disrupts asymmetric MTOC behavior in the following interphase. The data further indicate that neuroblasts are also sensitive to optogenetic manipulation of Cnb and Polo localization during interphase.

## Discussion

Centrosome asymmetry has previously been described to occur in asymmetrically dividing *Drosophila* neural stem cells (neuroblasts), manifested in biased interphase MTOC activity and asymmetric localization of the centrosomal proteins Cnb, Plp, and Polo [8,19–22] and PCM proteins like Centrosomin [9]. Here, we have shown that neuroblast centrosomes become intrinsically asymmetric by dynamically enriching centriolar proteins such as Cnb and Polo on the young daughter centriole during mitosis. This establishment of centriolar asymmetry is tightly linked to centriole-to-centrosome also called mitotic centriole conversion [28,31]. In early prophase, Cnb and Polo colocalize on the existing mother centriole of the apical centrosome but from late prometaphase onward, Cnb is exclusively and Polo predominantly localized on the daughter centriole. Mechanistically, these dynamic localization changes could entail a direct or indirect translocation of Cnb and Polo from the mother to the daughter centriole. This model is partially supported for Cnb with our FRAP data. Interestingly, Cnb behaves differently on the basal centrosome: The existing mother centriole does not contain any Cnb, appearing only on the forming daughter centriole in late prophase. This suggests a direct recruitment mechanism, which could also apply to the apical centrosome from anaphase onward. Our 3D-SIM, FRAP, and live-cell imaging data combined are most consistent with a model proposing that on the apical centrosome, a small pool of Cnb transfers from the mother to the daughter centriole during early mitosis. From anaphase onward, and from late prophase onward on the basal daughter centriole, Cnb levels increase through the recruitment of Cnb that was not previously associated with the mother centriole (Fig 8A and 8B).

Cnb is phosphorylated by the mitotic kinase Polo [19] and Polo-dependent phosphorylation of Cnb is necessary for its timely localization during mitosis. Interestingly, our data further suggest that Polo, which also becomes enriched on the daughter centriole during mitosis, is co-dependent with Cnb, while also requiring Wdr62. Polo's involvement in mitotic centriole conversion [31] further suggests that the same molecular machinery cooperatively converts a maturing centriole into a centrosome for the next cell cycle while simultaneously providing it with its unique molecular identity (Fig 8A–8C).

The mechanisms generating 2 molecularly distinct centrioles during mitosis seem to directly influence the centrosome's MTOC activity in interphase; the "Cnb$^+$, high Polo" daughter centriole will retain MTOC activity during interphase whereas the "Cnb$^-$, low Polo" mother centriole, separates from its daughter in early interphase and becomes inactive [19–22,42]. This model is in agreement with *bld10* or *plp* mutants, which fail to down-regulate Polo from the mother centriole, resulting in the formation of 2 active interphase MTOCs [21,22]. This is further supported by our mislocalization data, showing that optogenetic manipulation of Polo and Cnb asymmetry specifically during mitosis impacts MTOC activity in the subsequent interphase. However, we cannot exclude the possibility that MTOC asymmetry is also controlled independently of mitotic centrosome asymmetry establishment because optogenetic interphase manipulations of Polo and Cnb alone can also perturb biased MTOC activity.

Loss of Wdr62 or Cnb also affects asymmetric centriolar Polo localization. Yet, interphase centrosomes lose their activity in these mutants. *wdr62* mutants and *cnb* RNAi neuroblasts both show low Polo levels in interphase [20]. We thus hypothesize that in addition to an asymmetric distribution, Polo levels must remain at a certain level to maintain interphase MTOC activity; high symmetric Polo results in 2 active interphase MTOCs, whereas low symmetric Polo results in the formation of 2 inactive centrosomes. Indeed, our optogenetic experiment triggered an increase in centriolar Polo levels upon blue-light induction, suggesting that both Polo levels and distribution influence MTOC activity. This hypothesis is strengthened by Cnn's capacity to oligomerize and form a scaffold, supporting PCM assembly upon

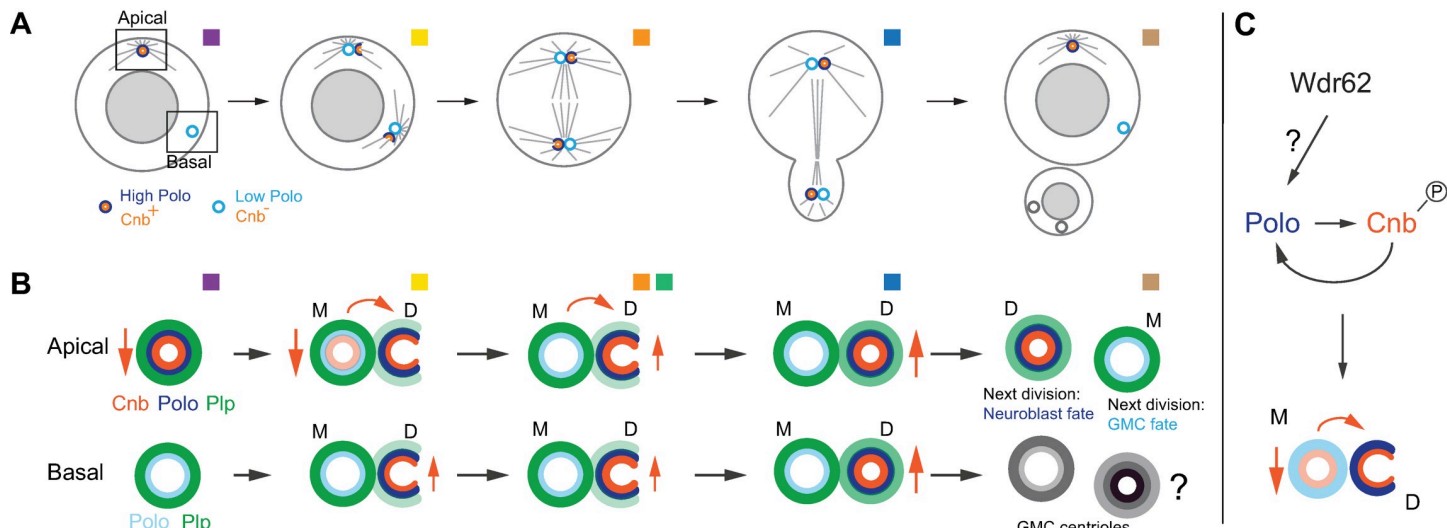

**Fig 8. Centrosome asymmetry is primed in mitosis through dynamic Cnb and Polo localization changes.** (A) Model: Centriolar asymmetry,—here shown for Polo (dark and light blue) and Cnb (orange),—occurs during mitosis, coupled to centriole-to-centrosome as known as mitotic centriole conversion. Polo and Cnb are enriched or restricted to the newly formed daughter centriole on both the apical and basal centrosome. The ensuing Cnb$^+$ and Polo-rich centriole maintains MTOC activity, retaining it in the self-renewed neuroblast. Details for the apical and basal centrosome are shown in panel B. Cnb (orange) and Polo (blue) localize dynamically on the forming daughter centriole and are down-regulated on the mother centriole from prophase onwards. The basal centrosome directly up-regulates Cnb on the daughter centriole where Polo is also enriched. Cnb's dynamic localization most likely entails both down- and up-regulation in prophase and prometaphase and up-regulation in anaphase and telophase, respectively (vertical orange arrows), as well as direct protein transfer (curved arrows). Plp (green) remains on the mother, potentially increasing in intensity and appearing on the daughter centriole in prometaphase. For the apical centrosome, the centriole containing less Plp, gained Cnb and Polo, is destined to be inherited by the self-renewed neuroblast (indicated with "neuroblast fate") in the next division, whereas the centriole containing higher Plp and lower Polo levels is destined to be inherited by the GMC (indicated with "GMC fate"). The fate of the basal centrioles and subsequent marker distribution is unknown (represented by gray circles). (C) Cnb and Polo co-depend on each other for their enrichment on the daughter centriole. Wdr62 is necessary for Polo's asymmetric centriolar localization, although the molecular mechanism is unclear. Cnb, Centrobin; GMC, ganglion mother cell; MTOC, microtubule organizing center; Plp, PCNT-like protein; Wdr62, WD40 repeat protein 62.

phosphorylation by Polo [43]. The more Polo is recruited, the more stable is the Cnn scaffold, supporting MTOC activity.

Centrobin is also enriched on the daughter centriole in mammals and preferentially becomes incorporated into the newly assembled daughter centriole in late G1 or early S phase. Centrobin remains localized at the daughter centrioles throughout the cell cycle [26]. It is of interest to note that in mammalian cells, Centrobin becomes enriched on daughter centrioles during the G1-S transition and not during centriole-to-centrosome conversion. It is tempting to speculate that other kinases and mechanism regulate this translocation.

We also observed that Cnb and Plp localize in a mutually exclusive manner, with Cnb localizing to the daughter centriole and Plp remaining on the mother centriole. In mammalian cells, similar mutual exclusion between the centriolar proteins Cep120 or Neurl4 and PCM proteins has been observed [24,44].

Taken together, the results reported here are consistent with a model, proposing that the establishment of 2 molecularly distinct centrioles is primed during mitosis and contributes to biased MTOC activity in the subsequent interphase. Wild-type neuroblasts unequally distribute a given pool of Cnb and Polo protein between the 2 centrioles so that the centriole inheriting high amounts of Cnb and Polo will retain MTOC activity. Furthermore, the dynamic localization of Polo and Cnb provides a molecular explanation for why the daughter-centriole–containing centrosome remains tethered to the apical neuroblast cortex and is being inherited by the self-renewed neuroblast [20–22] (Fig 8A). It remains to be tested why neuroblasts implemented such a robust machinery to asymmetrically segregate the daughter-

containing centriole to the self-renewed neuroblast. More refined molecular and behavioral assays will be necessary to elucidate the developmental and postdevelopmental consequences of biased centrosome segregation. The tools and findings reported here will be instrumental in targeted perturbations of intrinsic centrosome asymmetry with spatiotemporal precision in defined neuroblast lineages.

Finally, our observations reported here further raise the tantalizing possibility that centriolar proteins also dynamically localize in other stem cells, potentially providing a mechanistic explanation for the differences in centriole inheritance across different stem cell systems.

## Methods

### Fly strains, *trans*-genes and fluorescent markers

The following fly strains were used: Cnb RNAi (VDRC, 28651GD), $wdr62^{\Delta\,3-9}$ allele [20], *Df (2L)Exel8005* (a deficiency removing the entire *wdr62* locus and adjacent genes; BDSC #7779), *worniu-Gal4* [45], *pUbq-DSas-6::GFP* [46], *Cnn::GFP*, *Polo::GFP$^{CC01326}$* (protein trap line) [34], *GFP::Polo* (genomic rescue construct using Polo's endogenous enhancer) [40], *pUbq-Asl::GFP* [47], *pUbq-YFP::Cnb* [8], *YFP::Cnb$^{T4A,T9A,S82A}$* [8], *nos-Cas9/Cyo* (BDSC #78781), $y^1$, $w^{67c23}$, *P {y[+mDint2] = Crey}1b; D/TM3, Sb$^1$* (BDSC #851), $y^1$, *M{Act5C-Cas9.P.RFP-}ZH-2A*, $w^{1118}$, *DNAlig4$^{169}$* (BDSC #58492), *worGal4*, *UAS-mCherry::Jupiter* [16], *cnb$^{e00267}$* [19], *Df(3L) ED4284* (*cnb* deficiency; BDSC #8056), *polo$^1$* [48], *polo$^{16-1}$* [49], *pUASp-YFP::Cnb::PACT* [19].

The following mutant alleles and *trans*-genes were generated for this paper: *Polo::EGFP*, *SspB::EGFP::Polo*, *Plp::EGFP*, *Cnb::EGFP*, *SspB::Dendra2::Cnb*, *mCherry::Cnb::PACT*, *PACT:: HA::VhhGFP*, *SspB::mCherry*, *iLID::PACT::HA*, and *iLID::PACT::GFP*.

Unless specified otherwise, all strains were raised on standard medium at 25°C, under a 12L:12D light cycle.

### Generation of *trans*-genes

**CRISPR/Cas9 fusions.** Target specific sequences with high efficiency were chosen using the CRISPR Optimal Target Finder (http://tools.flycrispr.molbio.wisc.edu/targetFinder/), the DRSC CRISPR finder (http://www.flyrnai.org/crispr/), and the Efficiency Predictor (http://www.flyrnai.org/evaluateCrispr/) web tools. Sense and antisense primers for these chosen sites were then cloned into pU6-BbsI-ChiRNA [50] between BbsI sites.

Plp::EGFP Target Site 1: Sense: CTTCGAACTAGCGTCCACAAGGTC, Antisense: AAAC GACCTTGTGGACGCTAGTTC. Plp::EGFP Target Site 2: Sense: CTTCTGCTTATGGCTA CATTTGGG, Antisense: AAACCCCAAATGTAGCCATAAGCA. Polo::EGFP Target Site 1: Sense: CTTCGTCAGTCACCTCGGTGAATAT, Antisense: AAACATATTCACCGAGGTGA CTGAC. Polo::EGFP Target Site 2: Sense: CTTCGAGACTGTAGGTGACGCATTC, Antisense: AAACGAATGCGTCACCTACAGTCTC. Cnb::EGFP Target Site 1: Sense: CTTCG CTCTATGAGACCTAAGCCT, Antisense: AAACAGGCTTAGGTCTCATAGAGC. SspB:: EGFP::polo Target Site 1: Sense: CTTCGCTCTCCTTTCTTCTTTACTA, Antisense: AAAC TAGTAAAGAAGAAGGAGAGC. SspB::Dendra2::cnb Target Site 1: Sense: CTTCGGCAAC CCTGTGCATCACCA), Antisense: AAACTGGTGAT GCACAGGGTTGCC).

To generate the replacement donor template SspB [41] (Addgene #60416), the fluorophore (dendra2 or EGFP), and 1 kb homology arms flanking the insertion site were cloned into pHD-DsRed-attP (Addgene plasmid #51019) using Infusion technology (Takara/Clontech). Embryos expressing Act5C-Cas9 (BDSC#58492) for pHD-SspB::Dendra2::Cnb-DsRed, pHD-SspB::EGFP::Polo-DsRed or *nos-Cas9* [51] for Polo::EGFP, plp::EGFP and Cnb::EGFP were then injected with the replacement donor plasmid and its corresponding pU6-BbsI-ChiRNA. Injections were performed either in house or by Best Gene Injection Services

(www.thebestgene.com). Successful events were detected by DsRed-positive screening in the F1 generation. Constitutively active Cre (BDSC#851) was then crossed in to remove the DsRed marker. Positive events were then balanced, genotyped, and sequenced.

**Nanobody and optogenetic constructs.** PACT::HA::vhhGFP4: The coding sequences of PACT [37] and vhhGFP4 [38,39] were PCR amplified and cloned into a pUAST-attB vector using In-Fusion technology (Takara, Clontech). The HA sequence was then added using overhang PCR. The resulting construct was injected into attP flies for targeted insertion on third chromosome (VK00027, BestGene).

mCherry::Cnb::PACT: The coding sequences of mCherry and Cnb::PACT were amplified by PCR (Cnb::PACT was amplified from pUASp-YFP::Cnb::PACT [8]) and cloned into a pUASt-attB vector using In-Fusion technology (Takara, Clontech).The resulting construct was injected into attP flies for targeted insertion on the second chromosome (VK00018, BestGene).

SspB::mCherry: The coding sequence of SspB (Addgene #60416) and mCherry were PCR amplified and cloned into a pUAST-attB vector using In-Fusion technology (Takara, Clontech). An AgeI site was added in the primers sequences to be inserted between SspB and mCherry. The resulting construct was injected into attP flies for targeted insertion on the third chromosome (VK00033, BestGene).

UAS-iLID::PACT::HA: The coding sequence of iLID (addgene #60411) and PACT [37] were PCR amplified and cloned into a pUAST-attB vector using In-Fusion technology (Takara, Clontech) along with a synthesized HA oligonucleotide sequence. The resulting construct was injected into attP flies for targeted insertion on the second chromosome (VK00018, BestGene).

UAS-iLID::PACT::GFP: The coding sequence of iLID (addgene #60411), PACT [37], and GFP were PCR amplified and cloned into a pUAST-attB vector using In-Fusion technology (Takara, Clontech). An XhoI site was added in the primers sequences to be inserted between iLID and PACT, and an AgeI site was added between PACT and GFP. The resulting construct was injected into attP flies for targeted insertion on the third chromosome (VK00020, BestGene).

## Immunohistochemistry

The following antibodies were used for this study: rat anti-α-Tub (Serotec; 1:1,000), mouse anti-α-Tub (DM1A, Sigma; 1:2,500), rabbit anti-Asl (1:500), rabbit anti-Plp (1:1,000; gifts from J. Raff). Secondary antibodies were from Molecular Probes and the Jackson Immuno laboratory.

Ninety-six to one-hundred twenty hours after egg laying (AEL) larval brains were dissected in Schneider's medium (Sigma) and fixed for 20 min in 4% paraformaldehyde in PEM (100 mM PIPES [pH 6.9], 1 mM EGTA and 1 mM MgSO4). After fixing, the brains were washed with PBSBT (1X PBS, 0.1% Triton-X- 100, and 1% BSA) and then blocked with 1X PBSBT for 1 h. Primary antibody dilution was prepared in 1X PBSBT, and brains were incubated for up to 2 days at 4˚C. Brains were washed with 1X PBSBT 4 times for 20 minutes each and then incubated with secondary antibodies diluted in 1X PBSBT at 4˚C overnight. The next day, brains were washed with 1X PBST (1x PBS, 0.1% Triton-X- 100) 4 times for 20 minutes each and kept in Vectashield H-1000 (Vector laboratories) mounting media at 4˚C.

## Super-resolution 3D-SIM

3D-SIM was performed on fixed brain samples using a DeltaVision OMX-Blaze system (version 4; GE Healthcare), equipped with 405-, 445-, 488-, 514-, 568-, and 642-nm solid-state lasers. Images were acquired using a Plan Apo N 60x, 1.42 NA oil immersion objective lens

(Olympus) and 4 liquid-cooled sCMOs cameras (pco.edge 5.5, full frame 2560 × 2160; PCO). Exciting light was directed through a movable optical grating to generate a fine-striped interference pattern on the sample plane. The pattern was shifted laterally through 5 phases and 3 angular rotations of 60˚ for each z section. Optical z sections were separated by 0.125 μm. The laser lines 405, 488, 568, and 642 nm were used for 3D-SIM acquisition. Exposure times were typically between 10 and 120 ms, and the power of each laser was adjusted to achieve optimal intensities of between 5,000 and 8,000 counts in a raw image of 15-bit dynamic range at the lowest laser power possible to minimize photobleaching. Multichannel imaging was achieved through sequential acquisition of wavelengths by separate cameras.

### 3D-SIM image reconstruction

Raw 3D-SIM images were processed and reconstructed using the DeltaVision OMX SoftWoRx software package (version 6.1.3, GE Healthcare; Gustafsson, M. G. L. 2000). The resulting size of the reconstructed images was of 512 × 512 pixels from an initial set of 256 × 256 raw images. The channels were aligned in the image plane and around the optical axis using predetermined shifts as measured using a target lens and the SoftWoRx alignment tool. The channels were then carefully aligned using alignment parameter from control measurements with 0.5-μm diameter multispectral fluorescent beads (Invitrogen, Thermo Fisher Scientific).

### Live-cell imaging

Seventy-two to one-hundred twenty hours AEL larval brains were dissected in Schneider's medium (Sigma-Aldrich, S0146) supplemented with 10% BGS (HyClone) and transferred to 50 μL wells (Ibidi, μ-Slide Angiogenesis) for live-cell imaging. For Fig 6 and S4 Fig, live samples were imaged on a Perkin Elmer spinning disk confocal system "Ultra View VoX" with a Yokogawa spinning disc unit and 2 Hamamatsu C9100-50 frame transfer EMCCD cameras. A 63×/1.40 oil immersion objective mounted on a Leica DMI 6000B was used. Live-cell imaging data shown in Figs 2, 3 and 7 and S2 and S5 Figs was obtained with an Andor revolution spinning disc confocal system, consisting of a Yokogawa CSU-X1 spinning disc unit and 2 Andor iXon3 DU-897-BV EMCCD cameras. Either a 60×/1.4 NA or 100×/1.4 NA oil immersion objective mounted on a Nikon Eclipse Ti microscope was used.

### FRAP experiments

The 488-nm laser line was targeted to regions of interests using Andor's FRAPPA module. ROI's measured approximately 2 μm × 2 μm. Images were acquired every 30 to 60 s after bleaching event. EGFP intensity before and after bleaching was measured using Imaris' "Spot" tool. Background measurements were used to calculate the Signal/Noise (S/N), which was plotted in Fig 2C, 2E and 2F and S2C, S2D, S2G, S2I, S2J and S2K Fig.

### Optogenetics experiments

Crosses for optogenetics experiments were reared in the dark at 25˚C. Offspring from these crosses were raised in the dark and dissected after 4 days using red filters to minimize ambient and blue-light exposure. Optogenetic trapping or relocalization was performed using 10% to 20% of the 488-nm diode laser (50 mW) line.

### Centriolar age measurements

To determine centriolar age, Asl intensity was used as a reference. The contours of nonoverlapping centrioles were drawn in ImageJ based on Asl signal and saved as XY coordinates.

Using a custom-made MatLab code the total centriolar intensity, above background values determined by the experimenter, for Asl were calculated in the drawn centriolar areas. Total Asl intensity was then used to determine centriolar age because daughter centrioles have lower intensity than mother centrioles. The same XY coordinates were used to measure total pixel intensity for markers of interest (e.g., Polo::GFP, Plp::EGFP). Asymmetry ratios for markers of interest were then determined by dividing total daughter centriole intensity with total mother centriole intensity.

### Definition of statistical tests, sample number, sample collection, replicates

For each experiment, the data were collected from at least 2 independent experiments. All statistical details (replicates, $n$, statistical test used and $p$-values) for each experiment can be found in the corresponding figure legend. Statistical analyses were performed on Prism (GraphPad software). Statistical significance was assessed with a two-sided nonparametric Mann–Whitney test to compare ranks between 2 samples with variable variances and non-Gaussian distributions. $p < 0.05$ were considered significant; $^{*}p < 0.05$; $^{**}p < 0.01$; $^{***}p < 0.001$; $^{****}p < 0.0001$. A two-sided Fisher's exact test was used to examine the significance of the association between a phenotype (gain of MTOC activity, for instance) and a genotype. This test is adapted for all sample sizes. $p < 0.05$ were considered significant; $^{*}p < 0.05$; $^{**}p < 0.01$; $^{***}p < 0.001$; $^{****}p < 0.0001$.

### Code availability

Custom-made Matlab codes used for data analysis are available upon request.

### Supporting information

**S1 Fig. Centriole duplication completes during mitosis in larval neuroblasts.** (A) Representative 3D-SIM images of neuroblasts expressing the pericentriolar marker Cnn::GFP stained for α-Tubulin, labelling MTs (green). The morphology of the microtubule array and cell shape were used to define neuroblast cell-cycle stages. (B) Neuroblast centrosomes are inherently asymmetric in interphase but when neuroblast centrioles duplicate and acquire a unique molecular identity (indicated by arrow and color switch) is unknown. (C) Representative interpolated 3D-SIM images of third instar larval neuroblast centrosomes, expressing Sas-6::GFP (top row; white. Green in merge) and stained for Asl (middle row; white. Merged channels; magenta). The yellow arrowhead highlights the cartwheel of the forming centriole. Cartwheel duplication can be observed at the telophase to interphase transition, concomitantly with centrosome separation (blue arrowhead). Cell-cycle stages are indicated with colored boxes. Scale bar is 3 μm in panel A and 0.3 μm in panel C. GFP,; MT, microtubule; 3D-SIM, 3D structured illuminated microscopy.
(TIF)

**S2 Fig. Cnb is strongly recruited to the apical centrosome in late mitosis.** (A) Cnb enrichment at the end of mitosis could occur through new recruitment of cytoplasmic Cnb (vertical green arrows). (B) Representative image sequence and (C) intensity profile of a FRAPed wild-type neuroblast expressing endogenously tagged Cnb::EGFP (white; bottom row) together with the MT marker mCherry::Jupiter (white; top row). The orange brackets highlight the apical centrosome where Cnb::EGFP (bottom row) is measured. FRAPing occurred in late telophase. Colored vertical bars in the intensity plot indicate specific cell-cycle stages. The vertical dashed line refers to the time point when bleaching was performed. (D) Mean intensity plot of 8 FRAPed apical centrosomes in Cnb::EGFP expressing neuroblasts. Error bars indicate

standard deviation of the mean. Cnb intensity was plotted as a ratio of centrosomal Cnb/cytoplasmic Cnb (S/N; see Methods) in panels C and D. (E) Asl enrichment throughout mitosis could occur through constant loading of cytoplasmic Asl (magenta vertical arrows). Representative unFRAPed (F) and FRAPed (H) wild-type neuroblast expressing Asl::GFP (white; bottom row) together with the MT marker mCherry::Jupiter (white; top row). The orange brackets highlight the apical centrosome where Asl::GFP (bottom row) is measured. Intensity profile of the unFRAPed (G) and FRAPped (I) apical Asl::EGFP signal of the neuroblasts shown in panels F and H. Colored vertical bars indicate specific cell-cycle stages. The vertical dashed line refers to the time point when bleaching was performed. Mean intensity plot of 9 unFRAPed (J) and 8 FRAPed (K) apical centrosomes for neuroblasts expressing Asl::GFP. For panels G, I, J, and K, Asl intensity was plotted as a ratio of centrosomal Asl/cytoplasmic Asl (S/N; see Methods). Error bars indicate standard deviation of the mean. Time scale is mm:ss. Scale bar in panels B, F, and H is 5 μm (top row) and 1 μm (bottom row). The data presented here were obtained from 3 independent experiments. Numerical data for panels C, D, G, I, J, and K can be found in the file S1 Data.xlsx. Asl, Asterless; Cnb, Centrobin; EGFP, Enhanced green fluorescent protein; FRAP, fluorescence recovery after photobleaching; GFP, Green fluorescent protein; MT, microtubule; S/N, Signal/Noise.
(TIF)

**S3 Fig. Cnb's dynamic localization is controlled by Polo-dependent phosphorylation and Plp remains enriched on the mother centriole during mitosis.** Representative 3D-SIM images of the (A) first and (B) second centrosome of a *polo* mutant (*polo¹/polo¹⁶⁻¹*) third instar larval neuroblast expressing YFP::Cnb (white; middle row, green; bottom row) and stained for Asl (white; top row, magenta; bottom row). Yellow "D" and orange "M" refer to daughter and mother centrioles based on Asl intensity. *polo* mutant neuroblasts show a loss of MTOC activity during interphase, which randomizes centrosome positioning and distribution. Because we cannot distinguish between the "apical" and "basal" centrosomes anymore we refer to centrosome 1 and 2 instead; centrosome 1 being the most affected regarding Cnb mislocalization. Colored arrowheads and bars underneath the images highlight the degree of Cnb centriolar asymmetry. Graphs showing the timeline of Cnb's centriolar asymmetry at defined mitotic stages in (C) control (*polo/+*) and (D) *polo* mutant (*polo¹/polo¹⁶⁻¹*) neuroblasts. The bars show the percentage of neuroblasts containing a single Cnb⁺ centriole (dark blue), a single centriole without Cnb (dark gray), Cnb on both centrioles (transition stage with a daughter/mother ratio < 2; light green), predominant Cnb localization on the daughter centriole (strong asymmetry with a daughter/mother ratio between 2 and 10; light blue) or in which Cnb is completely shifted to the daughter centriole (complete asymmetry with a daughter/mother ratio > 10; light brown). This experiment was done 3 times. Representative 3D-SIM images of (E) apical and (F) basal third instar larval neuroblast centrosomes, expressing Plp::EGFP (white in middle row, green in merge), co-stained with Asl (white on top, magenta in merge). Orange and yellow shapes represent mother "M" and daughter "D" centrioles, respectively, based on Asl intensity. The number represents total Plp intensity ratios (daughter/mother centriole) in the shown image. Plp asymmetry ratios for the apical (red dots) and the basal (blue dots) centrosome are plotted in panel G from 3 independent experiments. Medians are shown in dark gray. Mann–Whitney test: Prometaphase: apical versus basal; *p* = 0.3856. Metaphase: apical versus basal; *p* = 0.2234. Anaphase: apical versus basal; *p* = 0.3583. Telophase: apical versus basal; *p* = 0.1844. Plp remains localized on the mother centriole on both centrosomes and enriches on the daughter centriole over time. Scale bar is 0.3 μm. Colored boxes indicate cell-cycle stages. Numerical data for panels C, D, and G can be found in the file S1 Data.xlsx. Asl, Asterless; Cnb, Centrobin; EGFP, enhanced green fluorescent protein; MTOC, microtubule

organizing center; Plp, PCNT-like protein; YFP, yellow fluorescent protein; 3D-SIM, 3D structured illuminated microscopy.
(TIF)

**S4 Fig. The PACT domain fused to the GFP-trapping nanobody perturbs centriolar asymmetry of GFP-tagged centrosomal proteins.** (A) Representative 3D-SIM images of third instar larval neuroblast centrosomes, expressing YFP::Cnb::PACT (white in the first row, green in the merge) and stained for Asl (magenta in the merge). The number represents total YFP::Cnb::PACT intensity ratios (daughter/mother centriole) in the shown image. YFP::Cnb:: PACT intensity ratios (daughter/mother centriole) are plotted in panel B. Yellow "D" and orange "M" refer to daughter and mother centrioles based on Asl intensity. (C) Representative live-cell imaging series from a neuroblast, recorded in the intact brain, expressing the microtubule marker mCherry::Jupiter (MTs, first row) and YFP::Cnb::PACT (second row). 3D-SIM and live-imaging experiments were performed 2 times each. (D) To test the function of centriolar asymmetry establishment, the asymmetric centriolar localization of Polo and Cnb needs to be perturbed. (E) The vhhGFP4 nanobody specifically traps GFP or YFP tagged proteins and was tethered to the mother centriole using Plp's PACT domain (F). Upon co-expression with GFP or YFP tagged centrosomal proteins in neuroblasts, centriolar asymmetry establishment was impaired (crossed-out arrows shown for Polo; blue). Representative live-cell image series from intact brains for neuroblasts expressing the microtubule marker mCherry::Jupiter (first row) and PACT::VhhGFP4 together with (G) YFP::Cnb, (I) Asl::GFP, and (K) GFP::Polo *trans*-gene (genomic rescue construct; see Methods). MTOC phenotype quantifications are shown for (H) YFP::Cnb, (J) Asl::GFP, and (L) GFP::Polo (blue; wild-type–like asymmetry, dark brown; loss of MTOC activity, light brown; gain of MTOC activity). Fisher's exact test: YFP::Cnb with and without PACT::VhhGFP4 expression; $p = 1.874 \times 10^{-14}$. Asl::GFP with and without PACT::VhhGFP4 expression; $p = 0.1751$. GFP::Polo with and without PACT::VhhGFP4 expression; $p = 1.105 \times 10^{-12}$. Cell-cycle stages are indicated with colored boxes. The data presented here were obtained from 2, 4, and 3 independent experiments for YFP::Cnb, Asl::GFP, and GFP::Polo, respectively. Time stamps are shown in hh:mm and "00:00" corresponds to telophase of the first division. Red and blue squares represent apical and basal centrosome, respectively (panels C, G, I, and K). Scale bar is 0.3 μm in panel A and 3 μm for the rest of the figure. Numerical data for panels B, H, J, and L can be found in the file S1 Data.xlsx. Cnb, Centrobin; Asl, Asterless; GFP, green fluorescent protein; MTOC, microtubule organizing center; PACT, pericentrin-AKAP-450 containing targeting; Plp, PCNT-like protein; YFP, yellow fluorescent protein; 3D-SIM, 3D structured illuminated microscopy.
(TIF)

**S5 Fig. Optogenetic protein recruitment is efficient on third instar larval neuroblast centrosomes.** (A) Representative time-lapse frames of a third instar neuroblast—imaged in an intact brain—expressing SspB::mCherry (second and third row; gray) together with iLID:: PACT::GFP (cyan; top row). Light exposure regime is indicated on the top. Orange brackets and red arrowheads highlight the apical neuroblast centrosome. An unrelated mCherry particle is highlighted with the green arrowhead. Intensity ratios, displaying the ratio of centrosomal/cytoplasmic SspB::mCherry are shown below; SspB::mCherry intensity was measured along a 12-pixel wide line covering the centriole and normalized against cytoplasmic mCherry levels. Note that SspB::mCherry is recruited from the cytoplasm to the apical centrosome within 5 seconds and released back into the cytoplasm within approximately 2 minutes. (B) Representative Prophase time-lapse frames of third instar larval neuroblasts expressing SspB:: EGFP::Polo alone (control; left) or in conjunction with iLID::PACT::HA (right). SspB::EGFP:: Polo (middle row; white) appears enriched and more focused in the presence of iLID::PACT::

HA and after blue-light exposure. Intensity ratios, displaying the ratio of centrosomal/cytoplasmic SspB::EGFP::Polo are shown below; SspB::EGFP::Polo intensity was measured along a 12-pixel wide line covering the centriole and normalized against cytoplasmic EGFP levels. Time scale is mm:ss. Scale bar is 1 μm for the third raw of panel A and 5 μm for other images. EGFP, Enhanced green fluorescent protein; GFP, green fluorescent protein; iLID, improved light-induced dimer; PACT, Pericentrin-AKAP-450 containing targeting; SspB, Stringent starvation protein B.
(TIF)

**S1 Movie. Wild type neuroblast expressing YFP::Cnb.** Wild-type control larval neuroblast expressing the centriolar marker YFP::Cnb (green) and the microtubule marker UAS-mCherry::Jupiter (white), driven by the neuroblast-specific worGal4 *trans*-gene. Note that the daughter centriole (Cnb$^+$) remains active and anchored to the apical cortex throughout interphase. The second centrosome matures in prophase (00:39) after it reached the basal side of the cell. "00:00" corresponds to telophase. Red and blue arrows indicate the neuroblast and GMC inherited centrosome, respectively. Note that the basal centrosome (blue) can be visualized based on its MTOC activity from prophase onwards. Time scale is hh:mm and the scale bar is 3 μm. Cnb, Centrobin; GMC, ganglion mother cell; MTOC, microtubule organizing center; YFP, yellow fluorescent protein.
(MP4)

**S2 Movie. Neuroblast expressing YFP::Cnb::PACT.** Larval neuroblast expressing YFP::Cnb::PACT (green) and the microtubule marker UAS-mCherry::Jupiter (white), driven by the neuroblast-specific worGal4 *trans*-gene. Note that YFP::Cnb::PACT is present on both centrioles. Both centrosomes remain active and anchored to the apical cortex throughout interphase. Centrioles split in prophase (00:39) accompanied by a large spindle rotation (00:42–00:45), resulting in normal asymmetric cell division. "00:00" corresponds to telophase. Red and blue arrows indicate the neuroblast and GMC inherited centrosome, respectively. Note that both centrosomes remain active and split only few minutes before mitosis. Time scale is hh:mm, and the scale bar is 3 μm. Cnb, Centrobin; GMC, ganglion mother cell; PACT, pericentrin-AKAP-450 containing targeting; YFP, yellow fluorescent protein.
(MP4)

**S3 Movie. Neuroblast expressing YFP::Cnb together with centriole tethered PACT::vhhGFP4.** Larval neuroblast expressing the centriolar marker YFP::Cnb (green), the microtubule marker UAS-mCherry::Jupiter (white) and the PACT::vhhGFP4 nanobody; both UAS lines are driven by the neuroblast-specific worGal4 *trans*-gene. The PACT domain confines the nanobody predominantly to the mother centriole. Both centrosomes remain active and anchored to the apical cortex throughout interphase. Centrosome splitting occurs a few minutes before mitosis (00:36). "00:00" corresponds to telophase. Red and blue arrows indicate the neuroblast and GMC inherited centrosome, respectively. Note that both centrosomes remain active and split only few minutes before mitosis. Time scale is hh:mm and the scale bar is 3 μm. Cnb, Centrobin; GMC, ganglion mother cell; PACT, pericentrin-AKAP-450 containing targeting; YFP, yellow fluorescent protein.
(MP4)

**S4 Movie. Neuroblast expressing Asl::GFP together with centriole tethered PACT::vhhGFP4.** Larval neuroblast expressing the centriolar marker Asl::GFP (green), the microtubule marker UAS-mCherry::Jupiter (white), and the PACT::vhhGFP4 nanobody; both UAS lines are driven by the neuroblast-specific worGal4 *trans*-gene. Similar to the wild-type control, the daughter centriole remains active and anchored to the apical cortex throughout

interphase. The mother centriole sheds its MTOC activity and moves away in early interphase (00:15). At mitotic entry (00:45), the mother centriole matures after it reached the basal side of the cell. "00:00" corresponds to telophase. Red and blue arrows indicate the neuroblast and GMC inherited centrosome, respectively. Note that the basal centrosome (blue) loses MTOC activity and split from the apical centrosome (red) shortly after mitosis and gains MTOC activity from prophase onwards. Time scale is hh:mm, and the scale bar is 3 μm. Asl, Asterless; GFP, Green fluorescent protein; GMC, ganglion mother cell; MTOC, microtubule organizing center; PACT, pericentrin-AKAP-450 containing targeting; UAS, upstream activation sequence. (MP4)

**S5 Movie. Wild-type control neuroblast expressing Polo::EGFP.** Wild-type control larval neuroblast expressing Polo::EGFP (green) engineered by CRISPR/Cas9 technology and the microtubule marker mCherry::Jupiter (white). Note that the daughter centriole remains active and anchored to the apical cortex throughout interphase. The mother centriole matures at 00:42 after it reached the basal cell cortex. "00:00" corresponds to telophase. Red and blue arrows indicate the neuroblast and GMC inherited centrosome, respectively. Note that the basal centrosome (blue) can be visualized based on its MTOC activity from prophase onwards. Time scale is hh:mm, and the scale bar is 3 μm. EGFP, enhanced green fluorescent protein; GMC, ganglion mother cell; MTOC, microtubule organizing center. (MP4)

**S6 Movie. Neuroblast expressing Polo::EGFP together with centriole tethered PACT:: vhhGFP4.** Larval neuroblast expressing Polo::EGFP (green) engineered by CRISPR/Cas9 technology, the microtubule marker mCherry::Jupiter (white), and the PACT::vhhGFP4 nanobody; both UAS lines are driven by the neuroblast-specific worGal4 *trans*-gene. Both MTOCs remain active and anchored to the apical cortex throughout interphase. Centrioles split only 6 minutes before mitosis starts (00:36). The mitotic spindle rotates significantly (00:42–00:48) to realign the spindle along the internal apical–basal polarity axis and to ensure normal asymmetric cell division. "00:00" corresponds to telophase. Red and blue arrows indicate the neuroblast and GMC inherited centrosome, respectively. Note that both centrosomes remain active and split only few minutes before mitosis. Time scale is hh:mm, and the scale bar is 3 μm. EGFP, enhanced green fluorescent protein; GMC, ganglion mother cell; MTOC, microtubule organizing cell; PACT, pericentrin-AKAP-450 containing targeting; UAS, upstream activation sequence. (MP4)

**S7 Movie. Neuroblast expressing GFP::Polo together with centriole tethered PACT:: vhhGFP4.** Larval neuroblast expressing the *trans*-gene GFP::Polo (green), the microtubule marker mCherry::Jupiter (white), and the PACT::vhhGFP4 nanobody; both UAS lines are driven by the neuroblast-specific worGal4 *trans*-gene. Both MTOCs remain active and anchored to the apical cortex throughout interphase. Centrioles split only 6 minutes before mitosis starts (00:48). "00:00" corresponds to telophase. Red and blue arrows indicate the neuroblast and GMC inherited centrosome, respectively. Note that both centrosomes remain active and split only few minutes before mitosis. Time scale is hh:mm, and the scale bar is 3 μm. GFP, green fluorescent protein; GMC, ganglion mother cell; MTOC, microtubule organizing ceneter; PACT, pericentrin-AKAP-450 containing targeting; UAS, upstream activation sequence. (MP4)

**S8 Movie. Neuroblast expressing SspB::EGFP::Polo together with centriole tethered iLID:: PACT::HA exposed to blue light during mitosis.** Larval neuroblast expressing SspB::EGFP::

Polo (green, right) engineered by CRISPR/Cas9 technology, the microtubule marker mCherry::Jupiter (white, left), and the iLID::PACT::HA optogenetic fusion protein both under the control of UAS, driven by the neuroblast-specific worGal4 *trans*-gene. Blue light is turned on during mitosis only as shown by Polo signal for the first 22 minutes. After light blue is turned off, both MTOCs remain active and anchored to the apical cortex throughout interphase. Active centrosomes split only 18 minutes before mitosis starts (1:23:15). "00:00" corresponds to the beginning of blue-light exposure, shortly before the first mitosis. Time scale is hh:mm:ss, and the scale bar is 3 μm. EGFP, enhanced green fluorescent protein; iLID, improved light-induced dimer; MTOC, microtubule organizing center; PACT, pericentrin-AKAP-450 containing targeting; SspB, Stringent starvation protein B; UAS, upstream activation sequence.
(MP4)

**S1 Data. Excel file containing all the numerical values relative to Fig 1E, 1F and 1G; Fig 2C, 2E and 2F; Fig 3B and 3G; Fig 4C; Fig 5C, 5E and 5F; Fig 6B, 6E, 6F, 6G, 6H, 6I and 6J; Fig 7D and 7E; S2C, S2D, S2G, S2I, S2J and S2K Fig; S3C, S3D and S3G Fig and S4B, S4H, S4J and S4L Fig.**
(XLSX)

## Acknowledgments

We thank members of the Cabernard lab for helpful discussions. We are grateful to Jordan Raff, Nasser Rusan, Tomer Avidor-Reiss, Cayetano Gonzalez, and Chris Doe for flies and antibodies. We would also like to thank the Imaging Core Facility (IMCF) at the Biozentrum for technical support and the Nigg and Affolter labs for providing temporary lab space to E. G.

## Author Contributions

**Conceptualization:** Emmanuel Gallaud, Anjana Ramdas Nair, Clemens Cabernard.

**Formal analysis:** Emmanuel Gallaud, Anjana Ramdas Nair, Clemens Cabernard.

**Funding acquisition:** Clemens Cabernard.

**Investigation:** Emmanuel Gallaud, Anjana Ramdas Nair, Nicole Horsley, Arnaud Monnard, Priyanka Singh.

**Methodology:** Emmanuel Gallaud, Anjana Ramdas Nair, Nicole Horsley, Arnaud Monnard, Priyanka Singh, David Salvador Garcia, Alexia Ferrand, Clemens Cabernard.

**Project administration:** Clemens Cabernard.

**Resources:** Emmanuel Gallaud, Arnaud Monnard, David Salvador Garcia, Clemens Cabernard.

**Software:** Tri Thanh Pham.

**Supervision:** Clemens Cabernard.

**Writing – original draft:** Clemens Cabernard.

**Writing – review & editing:** Emmanuel Gallaud, Clemens Cabernard.

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
