## [Editor Report · Decision Letter 0]

5 Mar 2020

Dear Dr Cabernard, 

Thank you for submitting your manuscript entitled "Dynamic centriolar localization of Polo and Centrobin in early mitosis primes centrosome asymmetry" for consideration as a Research Article by PLOS Biology.

Your manuscript has now been evaluated by the PLOS Biology editorial staff as well as by an academic editor with relevant expertise and I am writing to let you know that we would like to send your submission out for external peer review.

Please re-submit your manuscript within two working days, i.e. by Mar 09 2020 11:59PM.

Kind regards,

Ines

--

Ines Alvarez-Garcia, PhD

Senior Editor

PLOS Biology

Carlyle House, Carlyle Road

Cambridge, CB4 3DN

+44 1223–442810

---

## [Decision Letter · Decision Letter 1]

12 May 2020

Dear Dr Cabernard,

Thank you very much for submitting your manuscript entitled "Dynamic centriolar localization of Polo and Centrobin in early mitosis primes centrosome asymmetry" for consideration as a Research Article by PLOS Biology. Thank you also for your patience as we completed our editorial process, and please accept my apologies for the delay in providing you with our decision. As with all papers reviewed by the journal, yours was evaluated by the PLOS Biology editors as well as by an Academic Editor with relevant expertise and by three independent reviewers.

The reviews are attached below. As you will see, the reviewers are all very positive and think the results are important and significant for the field. Nevertheless, they ask for several clarifications that need to be addressed before we can consider the manuscript for publication.

Based on the reviews, we will probably accept this manuscript for publication, assuming that you will modify the manuscript to address the remaining points raised by the reviewers. Please also make sure to address the data and other policy-related requests noted at the end of this email.

We expect to receive your revised manuscript within two weeks, but please let us know if you need to request more time. Your revisions should address the specific points made by each reviewer. In addition to the remaining revisions and before we will be able to formally accept your manuscript and consider it "in press", we also need to ensure that your article conforms to our guidelines. A member of our team will be in touch shortly with a set of requests. As we can't proceed until these requirements are met, your swift response will help prevent delays to publication.

*Copyediting*

*Published Peer Review History*

*Early Version*

*Submitting Your Revision*

Sincerely,

Ines

--

Ines Alvarez-Garcia, PhD

Senior Editor

PLOS Biology

Carlyle House, Carlyle Road

Cambridge, CB4 3DN

+44 1223–442810

Fig. 1E, F, G; Fig. 2C, E, F; Fig. 3B, G; Fig. 4C; Fig. 5C, E; Fig. 6B, E, F; Fig. 7D, E; Fig. S2C, D, G, I, J, K; Fig. S3C, D, G and Fig. S4B, H, J, L

Reviewers’ comments

Rev. 1:

This study by Gallaud and colleagues investigates the mechanisms regulating centrosome asymmetry and biased centrosome segregation in Drosophila neural stem cells. Using a number of elegant techniques including genetic models, superresolution and live imaging, nanobody and optogenetic trapping, they demonstrate that the mitotic kinase Polo and its centriolar substrate Centrobin accumulate on the daughter centriole during mitosis, generating molecularly distinct mother and daughter centrioles. They discover that Centrobin's asymmetric localization may involve a direct re-localization mechanism, from mother to newly forming daughter centrioles, and is regulated by Polo-mediated phosphorylation. By manipulating the localization of these proteins, they propose that the establishment of centriole asymmetry in mitosis confers biased interphase centrosome activity, which is critical for correct spindle orientation.

Overall, the story is very intriguing, consistent with prior data, and the conclusions support the results. The work is very well performed, using multiple approaches to tackle each question. The data are presented clearly and are convincing. As such, I believe it is absolutely suitable for publication in PLoS Biology. I only have a few minor questions and concerns, which can be addressed by modifying the manuscript:

* Fig2 and page 10 - Based on the photobleaching experiments, the authors say that "In conclusion, we think that our data are most consistent with a hybrid model and propose that a small fraction of Cnb transfers from the mother to the daughter centriole in early mitosis through an unknown mechanism. From anaphase onward, the daughter centriole recruits additional Cnb from sources other than the mother centriole (Fig.2g)". Although this is a reasonable conclusion, an alternative interpretation would be that the reduction in Cnb levels on the mother centriole has nothing to do with its translocation to the daughter per se. Since the daughter centriole is templated off the wall of the mother centriole, one possibility is that the amount of Cnb in that (proximal) half of the mother centriole could be "displaced" as the cartwheel of the new procentriole forms. Furthermore, can the authors speculate as to why cells would need to go through this translocation? I'm assuming there is plenty of Cnb expressed and available to the forming procentriole… what would be the advantage of this translocation?!

* Fig 5B - what was the rationale for using a Cnb RNAi approach instead of the Cnb-phospho-mutant or hypomorphic strains?

* Fig 5 - The Cnb-PACT experiments are intriguing and show reversal of centriole asymmetries, at least with regards to Polo. Did the authors look at other centrosome maturation factors/PCM proteins to see if those are also reversed? I bring this up because in mammals, there appears to be an inverse relationship between asymmetrically localized, daughter centriole-enriched proteins (e.g. Cep120 or Neurl4) and the amount of PCM at that specific centriole (see for example Betleja et al, eLife 2018 or Li et al, EMBO Reports 2012). Thus, a mutual-exclusion model has been proposed. The authors may want to discuss those finding in the context of these data.

* Fig. 6 and 7 - I was a little bit confused about the outcomes of the experiments using nanobody trapping and optogenetic trapping. In the first case (Fig 6) the authors indicate that "preventing the normal establishment of Cnb and Polo asymmetry using the PACT domain perturbs biased MTOC activity in interphase", while in the latter (Fig 7) they suggest "perturbing normal Cnb and Polo asymmetry during mitosis disrupts asymmetric MTOC behavior in the following interphase". What I was waiting for was the phenotypic consequences of such manipulations! Was there a defect in segregation of cell fate markers? Did these manipulations disrupt asymmetric cell division? Were there any developmental defects associated with these, and are those consistent with what has previously been shown upon disruption of such processes?

* In the discussion, the authors do a very nice job of summarizing their results and putting them in context with previous work done in Drosophila. However, what is lacking is how these findings compare to the extensive data from mammalian cells with regards to asymmetrically localized proteins. I would highly encourage the authors to modify the Discussion section, remove some of the redundant summary of results, and add a paragraph to cover these instead. For example, there are a number of studies on Centrobin, Cep120 and Neurl4 that would warrant discussion here. Both Centrobin and Cep120 show daughter-centriole enrichment in mammals, and both show a similar loss from mother centrioles and enrichment on daughter centrioles coincident with procentriole assembly (e.g. Zou et al, JCB 2005; Mahjoub et al, JCB 2010). However, this occurs at the G1-S transition, which is NOT the stage at which centriole-to-centrosome conversion occurs. Thus, mechanisms for this translocation must exist outside of mitosis, and may be regulated by other kinases. Moreover, altering the localization of proteins such as Cep120 have been shown to impact accumulation of PCM proteins and microtubule dynamics in mitosis (e.g. Comartin et al, Current Biology, 2013) and in interphase (e.g. Betleja et al, eLife 2018). Thus, I think it is important to discuss the current results in the broader context. I believe the asymmetric regulation of centriolar and centrosomal proteins is an important topic; thus highlighting these studies and how they relate to what is known in the field in general would be very beneficial.

Rev. 2:

In this elegant paper the authors show that establishment of Polo and Centrobin (Cnb) asymmetry between engaged mother (low levels) and daughter (high levels) centrioles at the apical centrosome during mitosis in Drosophila neuroblasts is important for the establishment of centrosome asymmetry in the following interphase, which is itself necessary for subsequent spindle alignment. This is an important area of research as it directly relates to how asymmetric stem cell divisions are regulated. The authors use cutting edge super-res imaging and very powerful genetic tools to provide beautiful observations and establish their model. These tools will be of great value to others in the centrosome filed. The authors also reveal some mechanistic insight by providing good evidence that the asymmetry is established at least in part by a direct transfer of molecules from the mother to the daughter centriole. From looking at the pre-print version of this paper, it is clear this version of the paper has been improved by including the light-sensitive ectopic recruitment experiments that prove establishment of asymmetry during mitosis is necessary for proper asymmetry in the following interphase. The data is very clearly presented - I commend their Figure presentation, it is excellent and very helpful to the reader due to the complexity of the subject. I fully support publication in PloS Biology after the authors address my relatively minor concerns below.

1) "Cartwheels started to duplicate in late telophase, manifested in the appearance of a third Sas-6 positive cartwheel (blue arrowhead in Supplementary Fig.1c). Based on these data we conclude that in third instar larval neuroblasts centriolar cartwheels are duplicated in early interphase, forming a new procentriole. This procentriole subsequently converts into a mature centriole during mitosis through progressive loading of Asl. Thus, by the end of telophase, both neuroblast centrosomes contain two replication competent mature centrioles, an older mother and younger daughter centriole, which separate in the following interphase starting the cycle again." Did the authors always see that one centriole duplicated before the other in telophase (as seen in FigS1C)? If so, do they know which is destined to be the apical centrosome? I wonder in part because photoconversion results from Januschke et al., 2011 indicated that "at the time when apical and basal centrosomes split in Drosophila NBs, the non-motile apical centrosome contains only one centriole."

2) Authors should point out that while Polo and Cnb seem to be co-dependent, it is still possible to have Polo without Cnb because this is true of the mother centriole in the basal centrosome.

3) "Loss of Wdr62 or Cnb also affects asymmetric centriolar Polo localization. Yet, interphase centrosomes lose their activity in these mutants. wdr62 mutants and cnb RNAi neuroblasts both show low Polo levels in interphase (Ramdas Nair et al., 2016). We thus hypothesize that in addition to an asymmetric distribution, Polo levels must remain at a certain level to maintain interphase MTOC activity; high symmetric Polo results in two active interphase MTOCs whereas low symmetric Polo results in the formation of two inactive centrosomes. Indeed, our optogenetic experiment triggered an increase in centriolar Polo levels upon blue light induction, suggesting that both Polo levels and distribution influence MTOC activity." I think the link between low Polo levels and low MTOC activity can easily be explained by the findings that phosphorylation of Cnn by Polo causes Cnn to oligomerise into a scaffold that supports PCM assembly (Conduit et al., 2014, Dev Cell). This is likely also true at interphase apical centrosomes in neuroblasts. Indeed, it was shown that an asymmetry in Cnn occurs during early interphase and that Cnn is required for proper retention of the apical centrosome (Conduit and Raff 2010). The authors should discuss this.

4) "In early prophase neuroblasts, Polo was localized on the existing centriole on both centrosomes (Fig.4a,b & (Ramdas Nair et al., 2016))". Should this read "existing mother centriole"?

5) "Centrosome asymmetry has previously been described to occur in asymmetrically dividing Drosophila neural stem cells (neuroblasts), manifested in biased interphase MTOC activity and asymmetric localization of the centrosomal proteins Cnb, Plp and Polo (Lerit and Rusan, 2013; Ramdas Nair et al., 2016; Singh et al., 2014; Januschke et al., 2011; 2013)." I think here the authors should also cite Conduit and Raff 2010 and include Cnn in the list of proteins where asymmetry has been observed.

Rev. 3:

The study by Gallaud et al addresses the timing and mechanisms of establishment of an asymmetric identity between centrioles during the centrosomal cycle in drosophila, using the asymmetric division of larval neuroblasts, which are self-renewing stem cells in which the stereotyped orientation of the axis of division functionally relies on this asymmetry.

While the very mode of duplication of centrioles means that there is no ambiguity as to which is the mother and which is the daughter centriole, the question of how and when a daughter centriole acquires the molecular make up and identity of a mother is not entirely clear, nor is the dynamics of how this controls their differential behaviour. In the case of the dividing NB, this identity is manifest in the ability of one centriole to build an MTOC activity during the next cell cycle.

It was already known that larval NBs inherits the "youngest" centrosome from the previous cell cycle, and that on this centrosome, Centrobin (Cnb) localizes on the daughter centriole, which allows this centriole, after separation, to reconstitute an active MTOC activity, while its mother has no MTOC activity. It was also known that phosphorylation of Cnb by the polo kinase regulates its localization and activity. The current study addresses two questions:

- when does Cnb accumulate on the newly synthetized daughter centrioles, and disappear from the centriole that transits from a daughter to a mother identity.

- To which extent is the timing of this Cnb relocalization and accumulation at the time of mitosis important for the differential MTOC activity

The authors address these questions using a very complete and elegant arsenal of techniques, ranging from carefully timed super-resolution resolution imaging and characterization of centriolar molecular composition in wt and mutant conditions, to ectopic regulation of the localization of the main players via optogenetic approaches in order to study their temporal requirement for functional asymmetry.

Their main results/conclusions and advances are as follows:

1) A very detailed description of the dynamics of Cnb localization during the NB cell cycle. As the initially daughter/Cnb+/apical/MTOC+ centriole starts to duplicate upon mitotic entry, Cnb disappears from this centriole (that is transiting to a mother centriole) and almost simultaneously accumulates on the new daughter while it is being produced.

2) FRAP experiments suggest that at least part of Cbn on the new daughter derives from a relocalization of the pool that was initially present on its mother although an additional, newly synthetized pool of Cbn also participates in late mitosis. As the older mother/Cnb-/basal/MTOC- centriole duplicates to produce a new daughter, a new pool of Cnb accumulates on this new daughter. All these changes occur during mitosis

3) Phosphorylation of Cnb by polo contributes to (although apparently, at least in the genetic backgrounds used in the study, not 100% required for) its clearance from the "old" daughter, but is not required for its accumulation on the new daughters, which the authors summarize as "Polo dependent phosphorylation of Cnb is necessary for the establishment of molecularly distinct centrioles during mitosis, impacting subsequent molecular interphase asymmetry."

4) The localization of polo resembles that of Cnb, being cleared from the older apical centriole and accumulating on its daughter as it is formed during mitosis; it follows a similar dynamic on the basal centrosome. Its clearance from older and asymmetric accumulation on daughter centrioles reciprocally depends on Cnb;

5) Optogenetic ectopic recruitment of polo or Cbn to the mother centriole during mitosis is sufficient to prime an ectopic MTOC activity during the following interphase. However it is not necessarily required, as recruitment during interphase also results in high MTOC activity.

Although it is well written, the study is sometimes difficult to follow, as the distinctions between centriole and centrosome can be confusing, and it is not always immediately clear whether the terms mother and daughter refer to the previous or current cell cycle. But it is an inherent difficulty to this field, which requires very careful reading.

The functional approaches, based on gain of localization, are clever and beautifully performed. Hence one cannot help being slightly frustrated, as the authors may also feel, by some of the results that provide interesting hypotheses but are not entirely conclusive, such as those of the FRAP data or the optogenetic data.

Nonetheless, overall this is a beautifully detailed description of the dynamic change from daughter to mother centriole identity in the NB asymmetry context. The fine-grained temporal analyses, both in the purely descriptive data and for the functional approaches, provide a significant improvement over what had been shown in earlier studies on the same genes and proteins. Although the context is very specific, this study will in principle be of interest to a broad readership of cell biologists interested in the cell cycle, centrosome/centriole duplication, and more generally organelle reproduction; as well as to developmental/stem cell biologists with an interest in the molecular basis of centrosome asymmetry and its role in fate determination beyond the fly community.

Below are a few suggestions and some questions that should be addressed, and minor points

Concerning the set of optogenetics data: how stable is the ectopic recruitment of polo or Cbn to the mother centriole? The dissociation is only shown for a Cherry-tagged SspB (fig S5), but it is possible that although polo or Cbn are not normally present or recruited to the mother centrosome, an ectopic recruitment via the optogenetic approach would lead to a stable maintenance even after the stimulus is switched off. The interpretation of the "priming" of the strong MTOC activity would therefore be different, as in one case it would imply that the transient presence of polo or Cbn has stably induced a change in the mother centriole, whereas the other case would just mean that polo and/or cbn can directly induce the MTOC activity (as is suggested in Fig S5b, where the stronger recruitment of polo in prophase already induces a strong MTOC activity in the cell -judging on Jupiter-cherry signal), but not necessarily that they primed the centriole during mitosis for MTOC activity later on. Therefore, it would be important to monitor and describe the localization of SspB-polo and SspB-Cnb after optogenetic stimulation is stopped.

Along the same line, experiments in which Cnb or polo would be destroyed (eg using the AID degradation system or adaptation of the deGradFP system) specifically during mitosis or interphase would be more informative, although I do not know whether the temporal dynamics of such approaches is compatible with the short cycle time of the NBs

What happens to Cnb localization in PACt-vhhGFP4 + polo::EGFP? Is symmetric MTOC activity lost in PACt-vhhGFP4 + polo::EGFP in a Cnb-/- background?

Minor points:

In fig 6k, only the apical centrosome has strong levels of polo in ctrl cells and it is almost completely absent in the basal centrosome; However in figure 4b polo is present on the basal mother centriole in prophase and disappears in prometa. So is there a transient increase in prophase before it disappears?

S1C: Asl seems to display very similar intensities on both centrioles in the Sas6-GFP flies, whereas it is more asymmetric in the other experiments (eg Cnb-YFP). Could there be an artifact from using Sas6-GFP?

Fig3A-B: The quantified timeline displayed in 3B is a bit misleading. It is fixed material with no way to distinguish between apical/basal and old/young centrosome. So the dichotomy is established for each neuroblast and the centrosome whose mother centriole displays the highest amount of Cnb falls into the "strong delay category". As an obvious result, all centrosomes qualified as "high Cnb" ones will constitute a category displaying a stronger delay than the "low Cbn" one. This assumes an irreversible and coordinated progression (even if shifted in time) between the two centrosomes from mother with Cnb to mother without Cnb. This may be the case but should be explained.

Fig3C-G: these results are difficult to correlate with the 3D-SIM data. 3B show us that about 2/3 of cases display a normal "apical" type of Cnb distribution (adding centrosomes 1 and 2) while 3G, although at a later time point , shows an almost inverse 3/4 in favor of double Cnb cases. Please explain

Lines 75-76: very unclear phrasing…

Lines 145-7: "Cartwheels started to duplicate in late telophase, manifested in the appearance of a third Sas-6 positive cartwheel (blue arrowhead in Supplementary Fig.1c). Based on these data we conclude that in third instar larval neuroblasts centriolar cartwheels are duplicated in early interphase, forming a new procentriole." These conclusions are a little strong. Why do we see only one additional Sas6 dot and not two? How often do the authors see a Sas6 second dot on both centrioles and when does this second dot appears? Could the author give us an approximate duration of the interphase?

Line 232: "Since we cannot accurately distinguish between apical and basal centrosomes in cnb mutants expressing YFP::CnbT4A,T9A,S82A" A very brief reminder of the Cbn LOF could be useful to the reader.

Movies: for most of them the relevant centrioles/centrosomes are difficult to follow, please use some arrows or outlines

Fig7B: The diagram is clear at the individual level but it is difficult to understand the logic of the numerous different light activation regimes, with apparently very different illumination durations. A little clarification would be useful.

---

## [Editor Report · Decision Letter 2]

13 Jul 2020

Dear Dr Cabernard,

On behalf of my colleagues and the Academic Editor, Yukiko M Yamashita, I am pleased to inform you that we will be delighted to publish your Research Article in PLOS Biology. 

Early Version

PRESS 

Kind regards,

Alice Musson

Publishing Editor, 

PLOS Biology

on behalf of

Ines Alvarez-Garcia,

Senior Editor

PLOS Biology